# Plug-and-Play Guidance for Discrete Diffusion Models via Gradient-Informed Logit Correction

**Hongkun Dou** [1]   **Zike Chen** [1]   **Fengji Li** [1 2]   **Hongjue Li** [1]   **Yue Deng** [1 2 †]

## Abstract

Controllable generation with discrete diffusion models is often hindered by high computational overhead or the need for retraining. In this paper, we present **G**radient-**I**nformed **L**ogit **C**orrection (**GILC**), a plug-and-play framework that efficiently estimates guidance signals by repurposing the pretrained denoising network as a variational proxy. To circumvent the gradient instability inherent in high-dimensional discrete spaces, we introduce a Jacobian-free mechanism that directly corrects the clean prediction logits, facilitating stable and effective guidance. Our method accommodates both differentiable and non-differentiable reward functions. Extensive experiments across DNA, protein sequence, and molecular generation tasks demonstrate that GILC achieves state-of-the-art performance without additional training, frequently outperforming fine-tuning approaches.

Figure 1. **Illustration of the guided discrete diffusion process by GILC.** The reverse sampling process (top) iteratively denoises a DNA sequence from the fully masked state ($t = 1$) to the clean data ($t = 0$). The core correction mechanism (bottom) operates at each step: the mask predictor outputs the clean prediction $\mathbf{x}_\theta$, which are then modified by the reward gradient, $r(\cdot)$, yielding the guided prediction $\mathbf{x}_\theta^r$. The next state $\mathbf{z}_s$ is sampled from the transition distribution $p_\theta^r(\mathbf{z}_s|\mathbf{z}_t) = q(\mathbf{z}_s|\mathbf{z}_t, \mathbf{x}_\theta^r)$.

## 1. Introduction

Diffusion models (Ho et al., 2020; Song et al., 2021) have become the gold standard for probabilistic modeling in continuous domains, as powerfully demonstrated in image and video generation (Rombach et al., 2022; Esser et al., 2024; Liu et al., 2024). Correspondingly, recent studies (Austin et al., 2021; Lou et al., 2023; Shi et al., 2024; Sahoo et al., 2024) indicate that diffusion models also hold significant potential in discrete spaces, spanning domains such as language generation (Nie et al., 2025), biological sequence synthesis (Campbell et al., 2024), and molecular design (Vignac et al., 2023). While discrete diffusion effectively capture complex categorical data distributions, many scientific and industrial applications require generated samples that are not only realistic but also optimized for specific

task objectives, moving beyond unconditional generation. Examples include generating proteins with enhanced stability (Widatalla et al., 2024) or novel molecules with specific biochemical properties (Chang & Ye, 2024). Such tasks necessitate conditional or controllable generation, where classical approaches typically employ classifier/classifier-free guidance (Schiff et al., 2025; Nisonoff et al., 2025) or extensive model fine-tuning (Venkatraman et al., 2024; Rector-Brooks et al., 2025; Wang et al., 2025). However, these established methods demand additional training stages (e.g., training a separate classifier or retraining the generative model), which fundamentally limit their generalizability and introduce significant computational overhead.

An intriguing alternative is the plug-and-play paradigm, which aims to guide generation using off-the-shelf reward functions, neural predictors, or property evaluators without modifying or retraining the generative model (Chung et al., 2023; Yu et al., 2023; Song et al., 2023b; He et al., 2024). This flexible, training-free approach has achieved notable results in continuous diffusion. While conceptually

[1]Beihang University, Beijing, China [2]Zhongguancun Academy, Beijing, China. Correspondence to: Yue Deng <ydeng@buaa.edu.cn>.

*Proceedings of the 43rd International Conference on Machine Learning*, Seoul, South Korea. PMLR 306, 2026. Copyright 2026 by the author(s).

elegant, these methods struggle to directly translate to discrete diffusion due to the inherent non-differentiability of categorical data. Existing efforts based on Sequential Monte Carlo (SMC) (Wu et al., 2023; Uehara et al., 2024) or importance sampling (Li et al., 2024; Lin et al., 2025; Ou et al., 2026) often suffer from prohibitively high computational costs and demonstrably limited steering effectiveness. For a more comprehensive discussion of related work, we refer the reader to Appendix A.

This paper introduces **G**radient-**I**nformed **L**ogit **C**orrection (**GILC**), an efficient, principled, and training-free method for steering discrete diffusion (Fig. 1). We reframe the guidance task as estimating the gradient of a value function that measures the expected future reward from intermediate states. First, our framework employs a variational method where the pre-trained denoising network serves as a proxy for this value estimation. We then introduce two key practical insights for robust gradient computation: 1) combining the Gumbel-Softmax (GS) trick (Jang et al., 2017) and a Straight-Through (ST) estimator (Bengio et al., 2013) to maintain gradient flow in the discrete space, and 2) utilizing a Jacobian-free update that directly corrects the clean prediction logits for stable and effective guidance. Furthermore, we establish a formal connection between GILC and policy gradients (Williams, 1992) to universally handle non-differentiable objectives. Crucially, GILC operates entirely on off-the-shelf objective functions, requiring no fine-tuning or auxiliary training.

We demonstrate the effectiveness of GILC across a broad set of scientific domains, including DNA sequence design, protein sequence engineering, and multimodal molecular generation. Empirically, GILC not only significantly outperforms popular training-free discrete diffusion guidance methods in both sample quality and computational efficiency, but also competitively matches the performance of fine-tuning-based approaches, achieving state-of-the-art results for controlled discrete generation. These results highlight the immense potential of GILC for advancing the field of discrete diffusion guidance.

In summary, our contributions are as follows:

1. We propose a training-free guidance framework by demonstrating the pre-trained diffusion network's viability as a variational proxy for plug-and-play value function estimation.

2. We introduce logits correction guidance, which stably steers generation by computing the value gradient directly over the clean prediction logits, effectively resolving gradient instability issues in discrete spaces.

3. We establish a formal connection to the policy gradient formulation, extending the framework's universality to encompass non-differentiable objectives.

4. We empirically demonstrate state-of-the-art performance and computational efficiency for constrained generation across complex scientific domains.

## 2. Preliminaries

This section provides the foundational concepts for our work, covering the core principles of discrete diffusion models and the specific task formulation we address for constrained generation.

### 2.1. Discrete Diffusion Models

Diffusion models are a class of powerful generative models that define a forward process to progressively perturb data with noise, and a neural network is then trained to learn the reverse process to generate new data. In the context of discrete diffusion models, we operate on the data represented by one-hot encoded variables $\mathbf{x} \in \{0, 1\}^K$, where $K$ is the vocabulary size. These models use the categorical distribution, denoted as $\mathrm{Cat}(\cdot; \boldsymbol{\pi})$, to model the class probabilities, where $\boldsymbol{\pi} \in \Delta^K$ and $\Delta^K$ is the $K$-simplex.

While many discrete models define the forward process via a series of transition matrices (Austin et al., 2021; Lou et al., 2023), we adopt a simpler framework, proposed by Sahoo et al. (2024) and Shi et al. (2024), which directly interpolates between the clean data $\mathbf{x}$ and a fixed prior distribution $\boldsymbol{\pi}$:

$$q(\mathbf{z}_t|\mathbf{x}) = \mathrm{Cat}(\mathbf{z}_t; \alpha_t\mathbf{x} + (1 - \alpha_t)\boldsymbol{\pi}) \qquad (1)$$

where $\alpha_t$ is a predefined noise schedule that decreases from 1 as $t = 0$ to 0 as $t = 1$. This process implies a tractable transition probability $q(\mathbf{z}_t|\mathbf{z}_s) = \mathrm{Cat}(\mathbf{z}_t; \alpha_{t|s}\mathbf{z}_s + (1 - \alpha_{t|s})\boldsymbol{\pi})$ at each step, where $\alpha_{t|s} = \alpha_t/\alpha_s$. A key advantage of this design is that it yields a tractable posterior distribution $q(\mathbf{z}_s|\mathbf{z}_t, \mathbf{x})$, which is crucial for training and inference:

$$q(\mathbf{z}_s|\mathbf{z}_t, \mathbf{x}) = \mathrm{Cat}\left(\mathbf{z}_s; \frac{[\alpha_{t|s}\mathbf{z}_t + (1-\alpha_{t|s})\mathbf{1}\boldsymbol{\pi}^\top\mathbf{z}_t] \odot [\alpha_s\mathbf{x} + (1-\alpha_s)\boldsymbol{\pi}]}{\alpha_t\mathbf{z}_t^\top\mathbf{x} + (1-\alpha_t)\mathbf{z}_t^\top\boldsymbol{\pi}}\right) \qquad (2)$$

A particularly useful choice for the prior is an absorbing state, where all data eventually transitions to a single, special token (e.g., a [MASK] token). In this case, we set $\boldsymbol{\pi} = \mathbf{m}$, a one-hot vector corresponding to the mask token. With this choice, the posterior from the equation above simplifies into two distinct cases:

$$q(\mathbf{z}_s|\mathbf{z}_t, \mathbf{x}) = \begin{cases} \mathrm{Cat}(\mathbf{z}_s; \mathbf{z}_t) & \mathbf{z}_t \neq \mathbf{m} \\ \mathrm{Cat}\left(\mathbf{z}_s; \frac{(1-\alpha_s)\mathbf{m} + (\alpha_s - \alpha_t)\mathbf{x}}{1-\alpha_t}\right) & \mathbf{z}_t = \mathbf{m} \end{cases} \qquad (3)$$

The first case, $\mathbf{z}_t \neq \mathbf{m}$, shows a simple identity transition when the token is unmasked. The second case, $\mathbf{z}_t = \mathbf{m}$, demonstrates how the model can reverse the masking process to predict the original token. We use a neural network, parameterized by $\mathbf{x}_\theta(\mathbf{z}_t, t)$, to predict the clean data

and then substitute this prediction into the posterior, allowing us to perform the time-reversal sampling process via $p_\theta(\mathbf{z}_s|\mathbf{z}_t) = q(\mathbf{z}_s|\mathbf{z}_t, \mathbf{x}_\theta)$.

When processing a sequence of discrete tokens $\mathbf{x}^{1:L}$ (abbreviated as $\mathbf{x}$) of length $L$, the denoising process is assumed to be factorizable across tokens conditioned on the noisy sequence $\mathbf{z}_t^{1:L}$ (abbreviated as $\mathbf{z}_t$). This yields the full reverse process as a product of per-token distributions: $p_\theta(\mathbf{z}_s|\mathbf{z}_t) = \prod_{\ell=1}^{L} p_\theta(\mathbf{z}_s^\ell|\mathbf{z}_t)$. To train the network $\mathbf{x}_\theta$ to predict the original tokens for each position $\ell$, we minimize the following negative evidence lower bound (ELBO) objective:

$$\theta^* = \arg\min_\theta \mathbb{E}_{\mathbf{x}, \mathbf{z}_t} \int_0^1 \frac{\alpha_t'}{1-\alpha_t} \sum_{\ell=1}^{L} \log \langle \mathbf{x}_\theta^\ell(\mathbf{z}_t, t), \mathbf{x}^\ell \rangle \, \mathrm{d}t \tag{4}$$

Here, $\frac{\alpha_t'}{1-\alpha_t}$ is the weighting function, and the expectation is taken with respect to the forward process $q(\mathbf{z}_t|\mathbf{x})$. This forward process is assumed to be factorizable across tokens, conditioned only on the clean token at each position: $q(\mathbf{z}_t|\mathbf{x}) = \prod_{\ell=1}^{L} q(\mathbf{z}_t^\ell|\mathbf{x}^\ell)$. The term $\mathbf{x}_\theta^\ell(\mathbf{z}_t, t)$ represents the network's prediction of the clean token at position $\ell$, which is conditioned on the entire noisy sequence $\mathbf{z}_t$. Notably, this discrete diffusion formulation is equivalent to discrete flow models under the Continuous-Time Markov Chain (CTMC) framework (Campbell et al., 2024). Consequently, the proposed guidance method naturally extends to and can be directly applied within discrete flow models.

### 2.2. Task Formulation

We consider a scenario where we have a pre-trained discrete diffusion model, $p_\theta^{\mathrm{pre}}(\mathbf{x})$, trained using a standard objective like the ELBO. While such models excel at capturing the distribution of natural data, many real-world applications require generating data that satisfies specific properties.

Following prior work (Rafailov et al., 2023), we formulate this challenge as finding a new target distribution that balances adherence to a desired property with fidelity to the original pre-trained model. This is achieved by maximizing the following objective:

$$p_\theta^r(\mathbf{x}) = \arg\max_q \mathbb{E}_{\mathbf{x}\sim q} \underbrace{[r(\mathbf{x})]}_{\text{Reward}} - \beta \underbrace{\mathcal{D}_{\mathrm{KL}}\left[q(\mathbf{x})||p_\theta^{\mathrm{pre}}(\mathbf{x})\right]}_{\text{KL Regularization}} \tag{5}$$

The first term, the *Reward* $r(\cdot)$, quantifies how well a data sample $\mathbf{x}$ complies with the desired conditional constraints. This term can be modeled by an external function or an auxiliary network. The second term, *KL Regularization*, ensures that the new distribution remains close to the original pre-trained distribution $p_\theta^{\mathrm{pre}}(\mathbf{x})$. The hyperparameter $\beta$ controls the trade-off between satisfying the reward and maintaining the quality of the original gener-

ation. This objective has a well-known closed-form solution: $p_\theta^r(\mathbf{x}) \propto p_\theta^{\mathrm{pre}}(\mathbf{x}) \exp(r(\mathbf{x})/\beta)$ (Peters & Schaal, 2007; Peng et al., 2019).

The prevailing approaches to solving this objective typically involve fine-tuning the diffusion model or training a time-dependent classifier. However, these methods have significant drawbacks. Fine-tuning can suffer from reward hacking (Clark et al., 2024). Training a time-dependent classifier, while more flexible, still necessitates collecting new training data for each new target condition, which is both inconvenient and costly (Dhariwal & Nichol, 2021). In contrast to these established methods, our approach proposes a training-free strategy that can be seamlessly integrated with existing discrete diffusion and reward models. This allows for a flexible and efficient generation process without the need to modify model parameters or incur retraining.

## 3. Method

In this section, we propose **G**radient-**I**nformed **L**ogit **C**orrection (**GILC**), a general framework for guiding discrete diffusion in a plug-and-play fashion. We first demonstrate that solving the constrained generation objective (Eq. 5) relies on computing the gradient of a value function (Sec. 3.1). We then introduce a practical, training-free estimate for this value function via a variational proxy (Sec. 3.2). Finally, we present two robust approaches for calculating the necessary guidance gradient, including our key insight: omitting the Jacobian of the denoising network to achieve stable guidance by correcting the clean prediction logits (Sec. 3.3 and 3.4).

### 3.1. Taylor Series Approximation of the Optimal Reverse Process

The core idea behind guided generation is to replace the original, unguided reverse sampling step $p_\theta(\mathbf{z}_s|\mathbf{z}_t)$ with an optimal reverse process $p_\theta^r(\mathbf{z}_s|\mathbf{z}_t)$ that implicitly maximizes the target reward $r(\mathbf{x})$. As shown in prior work (Uehara et al., 2024; Li et al., 2024), this optimal process is given by the following formulation:

$$p_\theta^r(\mathbf{z}_s|\mathbf{z}_t) = \frac{p_\theta(\mathbf{z}_s|\mathbf{z}_t)\exp(v(\mathbf{z}_s)/\beta)}{\sum_{\mathbf{z}_s} p_\theta(\mathbf{z}_s|\mathbf{z}_t)\exp(v(\mathbf{z}_s)/\beta)} \tag{6}$$

where $v(\cdot)$ is the soft value function at state $\mathbf{z}_s$. It is rigorously defined as:

$$v(\mathbf{z}_s) = \beta \log \mathbb{E}_{p_\theta(\mathbf{x}|\mathbf{z}_s)}\left[\exp(r(\mathbf{x})/\beta)\right] \tag{7}$$

For practical implementation, especially when $\beta$ is small, this soft value function is commonly approximated by the expected reward: $v(\mathbf{z}_s) \approx \mathbb{E}_{p_\theta(\mathbf{x}|\mathbf{z}_s)}[r(\mathbf{x})]$. This approximation, justified in Li et al. (2024), simplifies the value function to the more intuitive concept of expected reward and helps mitigate numerical instability.

A significant challenge arises from the denominator in Eq. 6, which requires a summation over all possible states $\mathbf{z}_s$. For a sequence with length $L$, this involves a computationally intractable $K^L$ terms. In standard guidance frameworks (Schiff et al., 2025; Nisonoff et al., 2025), this obstacle is typically bypassed by training a parametric network $v_\phi(\cdot)$ to explicitly estimate the expected target reward:

$$\min_\phi \mathbb{E}_{q(\mathbf{z}_t|\mathbf{x})} \left(v_\phi\left(\mathbf{z}_t\right) - r(\mathbf{x})\right)^2 \qquad (8)$$

where $v_\phi(\cdot) : \mathbb{R}^{K \times L} \to \mathbb{R}$ is currently a continuously differentiable function and it optimal solution $v_\phi(\mathbf{z}_s) \approx \mathbb{E}_{p_\theta(\mathbf{x}|\mathbf{z}_s)}[r(\mathbf{x})]$. By treating discrete states as a point constrained within the continuous domain of the function (Grathwohl et al., 2021), we can apply a first-order Taylor series expansion to approximate $v_\phi(\mathbf{z}_s)$ around the current state $\mathbf{z}_t$:

$$
\begin{aligned}
v_\phi\left(\mathbf{z}_s\right) &\approx v_\phi\left(\mathbf{z}_t\right) + \left\langle \mathbf{z}_s - \mathbf{z}_t, \nabla_{\mathbf{z}_t} v_\phi(\mathbf{z}_t)\right\rangle \\
&= C + \sum_{\ell=1}^{L} \left\langle \mathbf{z}_s^\ell - \mathbf{z}_t^\ell, \nabla_{\mathbf{z}_t^\ell} v_\phi(\mathbf{z}_t)\right\rangle
\end{aligned} \qquad (9)
$$

The first term on the right-hand side is a constant with respect to $\mathbf{z}_s$ and can therefore be absorbed into the normalization constant of Eq. 6. The estimation of the optimal policy thus hinges entirely on calculating the value function gradient, which can be calculated efficiently with a single forward and backward pass of $v_\phi(\cdot)$. Moreover, the factorized form in the second line of Eq. 9 preserves the tractability of the reverse transition, enabling efficient sampling for high-dimensional discrete sequences. Additional details are provided in Appendix B.1.

Although this approach is reasonable, it requires retraining the value network $v_\phi(\cdot)$ for every new reward target. To overcome this constraint, the following section introduces a plug-and-play alternative that completely bypasses the retraining phase. Specifically, we demonstrate how this essential value gradient can be estimated directly by leveraging an off-the-shelf differentiable diffusion network. For notational brevity in subsequent derivations, we denote the differentiable value function $v_\phi(\cdot)$ simply as $v(\cdot)$.

### 3.2. A Variational Perspective on Value Function Estimation

As concluded in the previous subsection, the value function $v(\mathbf{z}_t)$ requires computing an expectation $\mathbb{E}_{p_\theta(\mathbf{x}|\mathbf{z}_t)}[\cdot]$ over the unrolled multi-step reverse process trajectory leading to the final data $\mathbf{x}$. Calculating or differentiating this expectation is computationally intractable. To overcome this, we introduce a variational approach by seeking a computationally efficient proxy distribution, $\tilde{p}(\mathbf{x}|\mathbf{z}_t)$, that closely approximates the true distribution $p_\theta(\mathbf{x}|\mathbf{z}_t)$. This is achieved

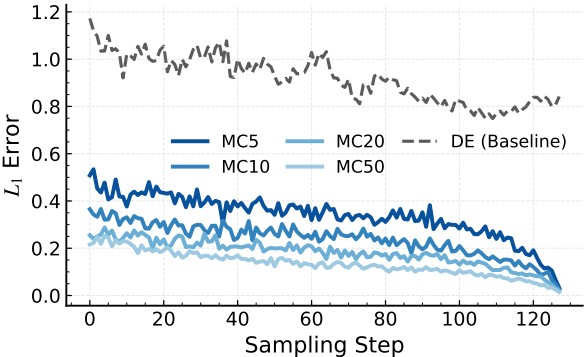

*(a)* Convergence of the value estimation error.

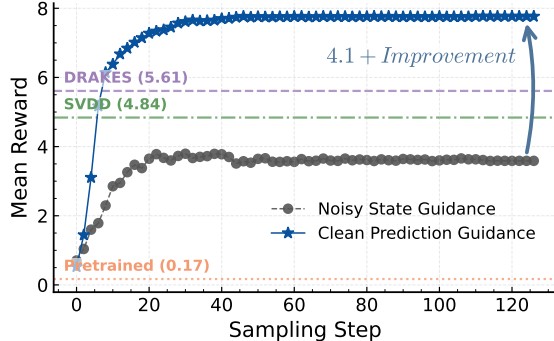

*(b)* Comparison of different guidance targets.

*Figure 2.* **Analysis of value estimation and guidance targets.** (a) Convergence of the $L_1$ error for the value function $v(\mathbf{z}_t)$ across varying Monte Carlo (MC) sample sizes. Increasing the sample size (from 5 to 50) consistently minimizes estimation error, significantly outperforming the Deterministic Estimation (DE) approach (Li et al., 2024). (b) Comparison of guidance targets during the denoising process. Guidance optimized in the clean-prediction logit ($\eta$) yields higher cumulative rewards and faster convergence compared to noisy-state guidance ($\mathbf{z}_t$). Results are reported on a discrete DNA diffusion model using GILC-DB.

by minimizing the KL divergence between the two:

$$
\begin{aligned}
&\arg\min_{\tilde{p}} \mathbb{E}_{\mathbf{x},\mathbf{z}_t}[\mathcal{D}_{\mathrm{KL}}(p_\theta(\mathbf{x}|\mathbf{z}_t)\|\tilde{p}(\mathbf{x}|\mathbf{z}_t))] \\
&\Leftrightarrow \arg\min_{\tilde{p}} \mathbb{E}_{\mathbf{x},\mathbf{z}_t}[-\log \tilde{p}(\mathbf{x}|\mathbf{z}_t)]
\end{aligned} \qquad (10)
$$

This equality shows that minimizing the KL divergence between our proxy and the true distribution is equivalent to minimizing the negative log-likelihood of the proxy. If we model our proxy using a mean-field assumption (Hoffman et al., 2013; Giordano et al., 2015), $\tilde{p}(\mathbf{x}|\mathbf{z}_t) = \prod_\ell \mathrm{Cat}(\mathbf{x}^\ell; \mu^\ell(\mathbf{z}_t, t))$, the objective becomes the minimization of the negative log-likelihood of the per-token predictions,

$$\arg\min_\mu \mathbb{E}_{\mathbf{x},\mathbf{z}_t} \sum_{\ell=1}^{L} -\log \left\langle \mu^\ell\left(\mathbf{z}_t, t\right), \mathbf{x}^\ell\right\rangle \qquad (11)$$

We observe that this objective is exactly the negative log-likelihood loss (Eq. 4) used to train the discrete diffusion

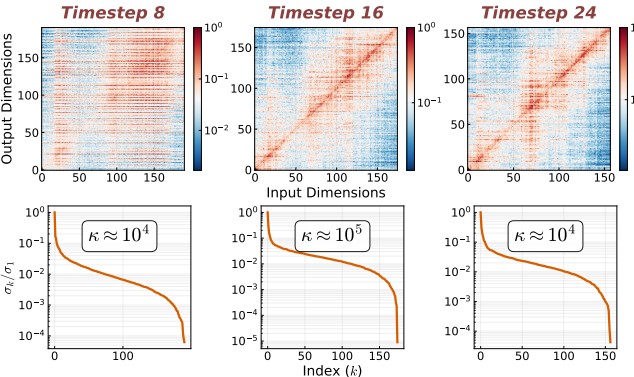

*Figure 3.* **Numerical instability in the model Jacobian.** (Top) Visualization of the Jacobian $\partial\eta/\partial\mathbf{z}_t$, aggregated over categorical dimensions using the Frobenius norm, for a discrete DNA diffusion model. (Bottom) Corresponding singular value spectrum. The high condition number ($\mathcal{K} \approx 10^4$–$10^5$) signifies severe ill-conditioning, which causes the gradient flow to become numerically unstable.

model's denoising network. This key insight means we do not need additional training and we can set the proxy distribution's parameters to the readily available predictions from our pre-trained model: $\mu\left(\mathbf{z}_t, t\right) \leftarrow \mathbf{x}_\theta\left(\mathbf{z}_t, t\right)$. This allows us to estimate the value function training-free using the Monte Carlo method through samples drawn from this variational proxy:

$$v\left(\mathbf{z}_t\right) \approx \frac{1}{n}\sum_{i=1}^{n} r(\mathbf{x}^{(i)}) \tag{12}$$

where samples $\mathbf{x}^{(1)}, \mathbf{x}^{(2)}, \cdots, \mathbf{x}^{(n)} \sim \tilde{p}(\mathbf{x}|\mathbf{z}_t)$.

***Comparison to Deterministic Estimation:*** Note that prior work (Li et al., 2024) has proposed a deterministic value estimation method by directly using the clean prediction input to the reward model, i.e., $v(\mathbf{z}_t) \approx r(\mathbf{x}_\theta\left(\mathbf{z}_t, t\right))$. In contrast, our approach utilizes stochastic sampling ($n > 1$) over the predicted distribution, which provides a better estimate of the expected future reward and captures the uncertainty inherent in the multi-step generation process.

As illustrated in Fig. 2a, we validate our approximation against the ground-truth value function, derived from full reverse rollout samples. Our empirical results demonstrate that deterministic methods suffer from significant estimation bias even in the final stages of sampling, resulting in a performance plateau. In contrast, our proposed estimator achieves superior precision that scales predictably with the number of samples, consistently narrowing the gap to the ground-truth value.

### 3.3. Practical Designs for Calculating Gradients

Our method requires calculating the gradient of the value function $\nabla_{\mathbf{z}_t} v(\mathbf{z}_t)$ to estimate the optimal reverse policy. This is achieved through two core designs that handle the

challenges of discrete sampling and network instability.

***1) Enabling Gradient Flow Through Discrete Sampling.*** The sampling operation required to estimate the value function, $v(\mathbf{z}_t) \approx \frac{1}{n}\sum r(\mathbf{x}^{(i)})$, breaks the computational graph, preventing direct differentiation. To restore the flow of gradients, we employ a combination of the Gumbel-Softmax (GS) reparameterization trick (Jang et al., 2017) and a Straight-Through (ST) estimator (Bengio et al., 2013).

The GS trick provides a differentiable, soft approximation of a categorical sample. Let $\eta = \eta(\mathbf{z}_t, t)$ be the prediction logits from the denoising network, where the clean prediction is $\mathbf{x}_\theta = \mathrm{Softmax}(\eta)$. The soft sample $\mathbf{x}_{\mathrm{soft}}$ is calculated as:

$$\mathbf{x}_{\mathrm{soft}} = \left[\frac{\exp((\eta_1+g_1)/\tau)}{\sum_{k=1}^{K}\exp((\eta_k+g_k)/\tau)}, \cdots, \frac{\exp((\eta_K+g_K)/\tau)}{\sum_{k=1}^{K}\exp((\eta_k+g_k)/\tau)}\right] \tag{13}$$

where $g_k \sim \mathrm{Gumbel}(0,1)$, and $\tau$ is a temperature coefficient controlling the sharpness of the approximation. However, many real-world reward models demand discrete or hard one-hot inputs. To ensure a consistent, accurate reward signal in the forward pass, we integrate the ST estimator. Specifically, we first obtain the hard, one-hot vector $\mathbf{x}_{\mathrm{hard}}$ by taking the argmax of the soft sample $\mathbf{x}_{\mathrm{hard}} = \mathrm{one\text{-}hot}(\arg\max_k(\mathbf{x}_{\mathrm{soft}}))$. We then compute the reward signal using the composite input $\hat{\mathbf{x}} = \mathbf{x}_{\mathrm{hard}} - \mathrm{sg}(\mathbf{x}_{\mathrm{soft}}) + \mathbf{x}_{\mathrm{soft}}$, where $\mathrm{sg}(\cdot)$ is the stop-gradient operation. This formulation ensures that the reward function processes the discrete $\mathbf{x}_{\mathrm{hard}}$ in the forward pass, while the gradient flows backward robustly through the differentiable $\mathbf{x}_{\mathrm{soft}}$.

***2) Logit Correction for the Clean Prediction.*** Following the reparameterization described above, the full gradient of the value function (Eq. 12) with respect to the noisy state $\mathbf{z}_t$ can be expanded via the chain rule:

$$\nabla_{\mathbf{z}_t} v(\mathbf{z}_t) \approx \frac{1}{n}\sum_{i=1}^{n} \underbrace{\frac{\partial r(\hat{\mathbf{x}}^{(i)})}{\partial \hat{\mathbf{x}}^{(i)}}\frac{\partial \hat{\mathbf{x}}^{(i)}}{\partial \eta}}_{\text{Logit Sensitivity}} \underbrace{\frac{\partial \eta}{\partial \mathbf{z}_t}}_{\text{Model Jacobian}} \tag{14}$$

While this formulation is mathematically rigorous, its direct computation is numerically unreliable. We observe that the model Jacobian $\partial\eta/\partial\mathbf{z}_t$ is often poorly conditioned (shown in Fig. 3), because denoising networks are trained to model clean data distributions, rather than to yield stable or smooth derivatives with respect to their inputs. As noted by Meng et al. (2021), a low training loss does not imply well-behaved Jacobian estimates. This instability is further exacerbated in discrete diffusion models, where the input $\mathbf{z}_t$ inhabits a discrete token space. Differentiating through this non-smooth structure introduces substantial noise into the gradient, preventing the guidance signal from accumulating coherently across the denoising trajectory, consistent with the empirical behavior observed in Fig. 2b.

To circumvent this bottleneck, we draw inspiration from Score Distillation Sampling (SDS) and its variants (Poole

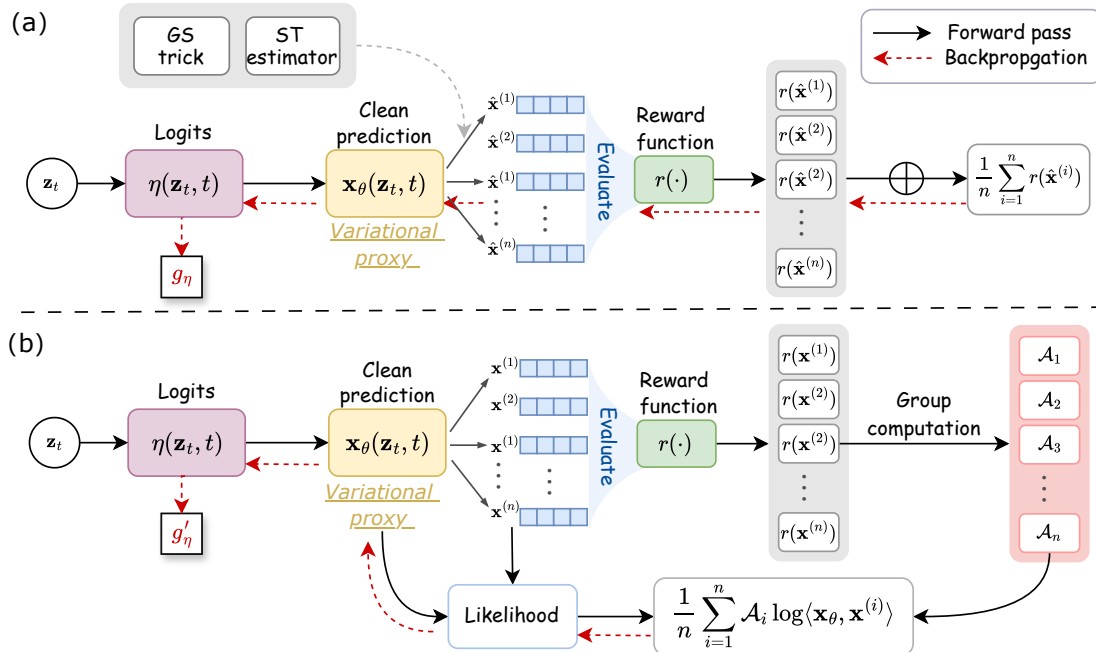

*Figure 4.* **Demonstration of gradient calculation for GILC.** (a) *Differentiable rewards:* The gradient $g_\eta$ is calculated via direct backpropagation, utilizing the Gumbel-Softmax trick and the Straight-Through estimator to enable differentiation through the discrete samples $\hat{\mathbf{x}}^{(i)}$. (b) *Non-differentiable rewards:* The gradient $g_\eta'$ is estimated via the policy gradient formulation, where the rewards $r(\mathbf{x}^{(i)})$ are converted into group relative advantages $\mathcal{A}_i$ to reduce variance. Both methods use the pre-trained network's clean prediction $\mathbf{x}_\theta$ as the variational proxy.

et al., 2023; Hertz et al., 2023), which omit unstable Jacobian terms in continuous diffusion. Analogously, we propose to bypass the model Jacobian altogether and define a stable guidance correction directly in the logit space:

$$g_\eta \triangleq \frac{1}{n} \sum_{i=1}^{n} \frac{\partial r(\hat{\mathbf{x}}^{(i)})}{\partial \hat{\mathbf{x}}^{(i)}} \frac{\partial \hat{\mathbf{x}}^{(i)}}{\partial \eta} \qquad (15)$$

This simplification is well-motivated: the clean-prediction logits $\eta$ lie in a smooth, continuous space where gradient propagation is numerically stable, whereas $\mathbf{z}_t$ resides in a discrete and highly non-smooth space. By extracting reward-induced perturbations directly in logit space, we preserve the informative gradient direction while avoiding the instability caused by differentiating through $\mathbf{z}_t$.

As shown in Appendix Appendix B.2, under this approximation the optimal reverse process $p_\theta^r(\mathbf{z}_s|\mathbf{z}_t)$ (Eq. 6) admits a particularly interpretable form. Specifically, the guidance first modifies the clean-prediction logits as $\eta^r = \eta + g_\eta/\beta$, yielding a reward-corrected prediction $\mathbf{x}_\theta^r = \text{softmax}(\eta^r)$. This corrected prediction is then incorporated into the reverse sampling step using the posterior from Eq. 3, namely $q(\mathbf{z}_s|\mathbf{z}_t, \mathbf{x}_\theta^r)$. We refer to the resulting algorithm as **G**radient-**I**nformed **L**ogit **C**orrection via **D**irect **B**ackpropagation (GILC-DB). A complete description is provided in Algorithm 1.

### 3.4. Guidance with Non-Differentiable Objectives

The gradient $g_\eta$ in Eq. 15 relies on the reward function $r(\cdot)$ being differentiable. While many rewards are provided by pre-trained differentiable networks, we propose an alternative, training-free approach for guiding systems that face a non-differentiable or black-box objective.

We address this using the concept of policy gradients (Williams, 1992). Our goal remains to calculate the gradient of the value function $v(\mathbf{z}_s) \approx \mathbb{E}_{p_\theta(\mathbf{x}|\mathbf{z}_s)}[r(\mathbf{x})]$ with respect to the prediction logits $\eta$. The derivation is as follows:

$$\begin{aligned} \nabla_\eta \mathbb{E}_{p_\theta(\mathbf{x}|\mathbf{z}_t)}[r(\mathbf{x})] &= \sum_{\mathbf{x}} \frac{\partial p_\theta(\mathbf{x}|\mathbf{z}_t)}{\partial \eta} r(\mathbf{x}) \\ &= \sum_{\mathbf{x}} p_\theta(\mathbf{x}|\mathbf{z}_t) \left( \frac{1}{p_\theta(\mathbf{x}|\mathbf{z}_t)} \frac{\partial p_\theta(\mathbf{x}|\mathbf{z}_t)}{\partial \eta} r(\mathbf{x}) \right) \\ &= \mathbb{E}_{p_\theta(\mathbf{x}|\mathbf{z}_t)} \left[ r(\mathbf{x}) \frac{\partial \log p_\theta(\mathbf{x}|\mathbf{z}_t)}{\partial \eta} \right] \end{aligned} \qquad (16)$$

In this expression, $p_\theta(\mathbf{x}|\mathbf{z}_t)$ acts as the policy, and the resulting expression facilitates estimation via Monte Carlo sampling. Recalling the insight from Sec. 3.2, we utilize the computationally tractable variational proxy $\tilde{p}(\mathbf{x}|\mathbf{z}_t)$ to draw a group of samples $\mathbf{x}^{(i)}$. The expectation is then approximated empirically, and the log-likelihood term $\log p_\theta(\mathbf{x}|\mathbf{z}_t)$ is replaced by the log-likelihood of the proxy, $\log \langle \mathbf{x}_\theta(\mathbf{z}_t, t), \mathbf{x}^{(i)} \rangle$.

To further enhance gradient stability and reduce variance, we shift our focus from absolute rewards to relative advantages, similar to Group Relative Policy Optimization (GRPO) (Shao et al., 2024). The resulting gradient for correcting the logits is expressed as:

$$g'_\eta \triangleq \frac{1}{n} \sum_{i=1}^{n} \mathcal{A}_i \frac{\partial \log \langle \mathbf{x}_\theta (\mathbf{z}_t, t), \mathbf{x}^{(i)} \rangle}{\partial \eta} \qquad (17)$$

where the advantage $\mathcal{A}_i = \frac{r(\mathbf{x}^{(i)}) - \text{mean}(\{r(\mathbf{x}^{(1)}), \cdots, r(\mathbf{x}^{(n)})\})}{\text{std}(\{r(\mathbf{x}^{(1)}), \cdots, r(\mathbf{x}^{(n)})\})}$ is computed using the rewards of the samples within the group. This formulation provides a robust, training-free mechanism for guidance that is entirely agnostic to the differentiability of the reward function, making it an efficient fallback for black-box guidance systems. We term this approach as **G**radient-**I**nformed **L**ogit **C**orrection via **P**olicy **G**radient (GILC-PG).

## 4. Experiments

In this section, we evaluate our guided generation methods, **GILC-DB** and **GILC-PG**, across three scientific domains: regulatory DNA sequence design, protein engineering, and small-molecule generation. In addition, we also conduct experiments on discrete diffusion in the image domain. A detailed ablation study of the key components is provided in Appendix D.

We compare our approach against the following categories. **1) Standard guidance methods.** Pretrained unconditional generation, Classifier Guidance (CG), and Classifier-Free Guidance (CFG) (Nisonoff et al., 2025). **2) Fine-tuning methods.** DRAKES (Wang et al., 2025), a specialized approach for discrete diffusion and flows, along with its variant without KL regularization (DRAKES w/o KL). **3) Training-free guidance methods.** Best-of-$N$ (Beirami et al., 2025); Sequential Monte Carlo (SMC) (Wu et al., 2023); SVDD (Li et al., 2024); and TFG-Flow (Lin et al., 2025). Implementation details and hyperparameter settings are provided in Appendix C.

### 4.1. Regulatory DNA Sequence Design

Here, our goal is to optimize regulatory DNA sequences to drive gene expression in a cell-type-specific manner (Taskiran et al., 2024).

***Experimental Setup.*** We conduct experiments on a large-scale enhancer dataset introduced by Gosai et al. (2023), which measures enhancer activity across human cell lines for approximately 700K DNA sequences of length 200 bp using massively parallel reporter assays (MPRAs). For each sequence, the corresponding gene expression level is provided. We employ a masked discrete diffusion model (Sahoo et al., 2024) pretrained on the full sequence set. In addition,

we use two reward oracles trained on disjoint subsets of the data: one used during guided generation and the other reserved exclusively for evaluation. Both oracles adopt the Enformer architecture (Avsec et al., 2021) to predict enhancer activity in the HepG2 cell line.

To comprehensively evaluate enhancer generation performance, we generate 640 DNA samples and report metrics spanning three complementary aspects, following prior work (Wang et al., 2025): 1) *Functional activity*, including predicted activity (Pred-Activity) and binary classification of chromatin accessibility (ATAC-Acc); 2) *Sequence fidelity*, measuring similarity to real enhancer sequences via 3-mer Pearson correlation (3-mer Corr) and JASPAR motif enrichment correlation (JASPAR Corr); and 3) *Distributional consistency*, which assesses whether generated sequences remain within the data manifold using approximated log-likelihood (App-Log-Lik).

***Results.*** Tab. 1 reports the quantitative results. Sequences generated by GILC-DB achieve the highest predicted activity in HepG2 cell lines. In particular, its Pred-Activity and ATAC-Acc metrics substantially outperform all other training-free baselines, and even surpass fine-tuned methods such as DRAKES. GILC-PG demonstrates superior similarity to natural enhancers, as evidenced by strong correlations with trinucleotide distributions and JASPAR motif patterns, together with favorable likelihood scores. In addition, Tab. 2 compares the computational efficiency of different methods. Best-of-$N$, SMC, and SVDD require maintaining multiple sampling trajectories, whereas the proposed GILC methods operate with a single trajectory. Notably, GILC-DB explicitly exploits the differentiability of the reward function $r(\cdot)$, resulting in significantly fewer reward evaluations than competing approaches.

### 4.2. Protein Sequence Design

Given a pre-trained inverse folding model that generates protein sequences conditioned on a fixed 3D backbone conformation, our objective is to guide the generation process toward sequences with enhanced thermodynamic stability.

***Experimental Setup.*** We utilize a pretrained discrete flow model (Campbell et al., 2024), using the ProteinMPNN architecture (Dauparas et al., 2022) as its backbone. The model is pretrained for inverse folding on the PDB dataset curated by Dauparas et al. (2022). To guide the generation process, we employ the reward oracle trained on the Megascale protein stability dataset (Tsuboyama et al., 2023), which comprises experimental stability measurements for approximately 1.8 million sequence variants across 983 protein domains. We maintain two distinct ProteinMPNN-based oracles: a guidance oracle used during sampling and an independent evaluation oracle reserved for final validation. Both oracles are trained to predict protein stability.

*Table 1.* Performance of various methods in regulating DNA sequence design. The table reports the mean and standard deviation of three random seeds. **Bold**: best, underline: second best.

| Method | Pred-Activity ↑ | ATAC-Acc ↑ (%) | 3-mer Corr ↑ | JASPAR Corr ↑ | App-Log-Lik ↑ |
|---|---|---|---|---|---|
| Pretrained | $0.17_{\pm 0.04}$ | $1.5_{\pm 0.2}$ | $-0.061_{\pm 0.034}$ | $0.249_{\pm 0.015}$ | $\underline{-261}_{\pm 0.6}$ |
| CG (Nisonoff et al., 2025) | $3.30_{\pm 0.00}$ | $0.0_{\pm 0.0}$ | $-0.065_{\pm 0.001}$ | $0.212_{\pm 0.035}$ | $-266_{\pm 0.6}$ |
| CFG (Nisonoff et al., 2025) | $5.04_{\pm 0.06}$ | $92.1_{\pm 0.9}$ | $0.746_{\pm 0.001}$ | $0.864_{\pm 0.011}$ | $-265_{\pm 0.6}$ |
| DRAKES w/o KL (Wang et al., 2025) | $\underline{6.44}_{\pm 0.04}$ | $82.5_{\pm 2.8}$ | $0.307_{\pm 0.001}$ | $0.557_{\pm 0.015}$ | $-281_{\pm 0.6}$ |
| DRAKES (Wang et al., 2025) | $5.61_{\pm 0.07}$ | $\underline{92.5}_{\pm 0.6}$ | $0.887_{\pm 0.002}$ | $0.911_{\pm 0.002}$ | $-264_{\pm 0.6}$ |
| Best-of-$N$ (Beirami et al., 2025) | $3.73_{\pm 0.41}$ | $35.8_{\pm 7.6}$ | $0.813_{\pm 0.037}$ | $0.671_{\pm 0.071}$ | $-262_{\pm 1.1}$ |
| SMC (Wu et al., 2023) | $4.15_{\pm 0.33}$ | $39.9_{\pm 8.7}$ | $0.840_{\pm 0.045}$ | $0.756_{\pm 0.068}$ | $\textbf{-259}_{\pm 2.5}$ |
| SVDD (Li et al., 2024) | $4.84_{\pm 0.28}$ | $51.9_{\pm 6.3}$ | $0.870_{\pm 0.061}$ | $0.826_{\pm 0.057}$ | $-269_{\pm 3.1}$ |
| TFG-Flow (Lin et al., 2025) | $3.48_{\pm 0.21}$ | $21.2_{\pm 5.4}$ | $0.262_{\pm 0.058}$ | $0.566_{\pm 0.044}$ | $-263_{\pm 1.5}$ |
| **GILC-DB (Ours)** | $\textbf{7.04}_{\pm 0.26}$ | $\textbf{95.2}_{\pm 2.1}$ | $\underline{0.900}_{\pm 0.044}$ | $\underline{0.935}_{\pm 0.039}$ | $-267_{\pm 1.8}$ |
| **GILC-PG (Ours)** | $5.21_{\pm 0.18}$ | $84.0_{\pm 1.5}$ | $\textbf{0.910}_{\pm 0.027}$ | $\textbf{0.937}_{\pm 0.031}$ | $-270_{\pm 2.4}$ |

*Table 2.* Comparison of the number of calls to the denoising model and the reward function per step across different training-free guidance approaches in the DNA sequence design task.

| Method | Num-of-Diff ↓ | Num-of-Reward ↓ |
|---|---|---|
| Best-of-$N$ | 20 | 20 |
| SMC | 20 | 20 |
| SVDD | 20 | 20 |
| TFG-Flow | 1 | 20 |
| **GILC-DB (Ours)** | 1 | 5 |
| **GILC-PG (Ours)** | 1 | 20 |

We employ the following three types of metrics to evaluate the stability of generated sequences and their ability to fold into target structures. 1) *Predicted stability*, evaluated using the independent evaluation oracle (Pred-ddG); 2) *Self-consistency*, measures the structural fidelity of the generated sequences. We utilize ESMFold (Lin et al., 2023) to predict the folded structures and calculate the Root Mean Square Deviation (RMSD) relative to the wild-type structure (scRMSD). 3) Finally, we define the *success rate* as the percentage of sequences that simultaneously satisfy Pred-ddG $> 0$ and scRMSD $< 2$ as in Campbell et al. (2024).

***Results.*** As shown in Tab. 3, GILC-DB generates protein sequences with high structural stability, achieving the highest Pred-ddG scores while maintaining inverse folding success rates comparable to those of pre-trained models, measured by the percentage of samples with scRMSD $< 2$. When these criteria are considered jointly, GILC-DB substantially outperforms all baseline methods in terms of overall success rate, exceeding DRAKES by approximately 4 percentage points. GILC-PG also consistently outperforms other training-free baselines. By contrast, DRAKES is susceptible to reward hacking (Clark et al., 2024) and requires careful tuning of the KL-constraint strength, while CFG relies heavily on labeled data and consequently suffers from limited generalization. Overall, these results highlight the strong potential of GILC for protein sequence design.

### 4.3. Multimodal Molecule Generation

We evaluate our method on the inverse design of molecules with targeted properties, a core challenge in computational chemistry (Hoogeboom et al., 2022; Gebauer et al., 2022).

***Experimental Setup.*** Following Lin et al. (2023), we adopt a pre-trained multimodal flow model trained on QM9 (Ramakrishnan et al., 2014) with an Equivariant Graph Neural Network (EGNN) backbone (Satorras et al., 2021), which jointly models discrete atomic types and continuous 3D coordinates. To avoid information leakage, the dataset is split into two disjoint subsets: one for training the guidance (reward) predictor and the other for training an independent evaluation predictor.

We apply GILC and training-free baselines to guide discrete atomic types toward following quantum properties: polarizability ($\alpha$), dipole moment ($\mu$), heat capacity ($C_v$), HOMO energy ($\epsilon_{\text{HOMO}}$), LUMO energy ($\epsilon_{\text{LUMO}}$), and energy gap ($\Delta\epsilon$). For each property, 4,096 samples are generated for evaluation. Guidance performance is assessed using mean absolute error (MAE) with the evaluation predictor.

***Results.*** Quantitative results summarized in Tab. 4. Both GILC-DB and GILC-PG consistently outperform all unsupervised baselines, including TFG-FLOW and SVDD. GILC-DB achieves the best overall performance by exploiting intrinsic reward gradients for more accurate guidance. Conversely, GILC-PG remained highly competitive despite treating the attribute predictor as a black box, offering superior flexibility. These results demonstrate the effectiveness and generality of the GILC framework for multimodal scientific data generation.

### 4.4. Discrete image generation

***Experimental Setup.*** To evaluate the scalability and versatility of the GILC framework in larger discrete spaces, we conduct class-conditional and text-to-image generation

*Table 3.* Model performance in the inverse protein folding task. The table reports the mean and standard deviation of three random seeds. **Bold**: best, underline: second best. The primary metric (overall success rate) is further highlighted in **blue**.

| Method | Pred-ddG $\uparrow$ | %(ddG $> 0$) (%)$\uparrow$ | scRMSD $\downarrow$ | %(scRMSD$< 2$)(%)$\uparrow$ | Success Rate (%)$\uparrow$ |
|---|---|---|---|---|---|
| Pretrained | $-0.544_{\pm 0.037}$ | $36.6_{\pm 1.0}$ | $0.849_{\pm 0.013}$ | $90.9_{\pm 0.6}$ | $34.4_{\pm 0.5}$ |
| CG (Nisonoff et al., 2025) | $-0.561_{\pm 0.045}$ | $36.9_{\pm 1.1}$ | $0.839_{\pm 0.012}$ | $90.9_{\pm 0.6}$ | $34.7_{\pm 0.9}$ |
| CFG (Nisonoff et al., 2025) | $-1.186_{\pm 0.035}$ | $11.0_{\pm 0.4}$ | $3.146_{\pm 0.062}$ | $29.4_{\pm 1.0}$ | $1.3_{\pm 0.4}$ |
| DRAKES w/o KL (Wang et al., 2025) | $\underline{1.108}_{\pm 0.004}$ | $\mathbf{100.0}_{\pm 0.0}$ | $7.307_{\pm 0.054}$ | $34.1_{\pm 0.2}$ | $34.1_{\pm 0.2}$ |
| DRAKES (Wang et al., 2025) | $1.095_{\pm 0.026}$ | $86.4_{\pm 0.2}$ | $0.918_{\pm 0.006}$ | $91.8_{\pm 0.5}$ | $\underline{78.6}_{\pm 0.7}$ |
| Best-of-$N$ (Beirami et al., 2025) | $0.623_{\pm 0.051}$ | $45.5_{\pm 4.7}$ | $0.849_{\pm 0.011}$ | $91.2_{\pm 0.5}$ | $54.7_{\pm 5.3}$ |
| SMC (Wu et al., 2023) | $0.659_{\pm 0.044}$ | $68.5_{\pm 3.1}$ | $0.841_{\pm 0.006}$ | $\mathbf{93.8}_{\pm 0.4}$ | $63.6_{\pm 4.0}$ |
| SVDD (Li et al., 2024) | $0.694_{\pm 0.076}$ | $69.3_{\pm 2.1}$ | $0.850_{\pm 0.030}$ | $89.7_{\pm 0.9}$ | $65.0_{\pm 3.3}$ |
| TFG-Flow (Lin et al., 2025) | $0.410_{\pm 0.064}$ | $39.2_{\pm 1.1}$ | $\mathbf{0.837}_{\pm 0.021}$ | $92.6_{\pm 1.6}$ | $52.9_{\pm 2.2}$ |
| **GILC-DB (Ours)** | $\mathbf{1.430}_{\pm 0.073}$ | $\underline{97.9}_{\pm 1.9}$ | $0.968_{\pm 0.015}$ | $84.3_{\pm 2.0}$ | $\mathbf{82.4}_{\pm 2.5}$ |
| **GILC-PG (Ours)** | $0.719_{\pm 0.091}$ | $75.6_{\pm 3.5}$ | $0.914_{\pm 0.012}$ | $92.4_{\pm 2.3}$ | $69.8_{\pm 3.2}$ |

*Table 4.* Mean absolute error (MAE) of generated target quantum properties on the QM9 dataset, evaluated using the property predictor. Upper-bound, #Atoms, and lower-bound results are taken from the study (Bao et al., 2023). Mean and standard deviation are calculated over three random seeds. Among training-free methods, **bold** indicates the best performance, and underline indicates the second best.

| Method | $C_v$ | $\alpha$ | $\mu$ | $\Delta\epsilon$ | $\epsilon_{\mathrm{HOMO}}$ | $\epsilon_{\mathrm{LUMO}}$ |
|---|---|---|---|---|---|---|
| Upper bound | 6.87 | 1.61 | 8.98 | 1464 | 645 | 1457 |
| #Atoms | 1.97 | 1.05 | 3.86 | 886 | 426 | 813 |
| Lower bound | 0.040 | 0.043 | 0.09 | 65 | 39 | 36 |
| Best-of-$N$ (Beirami et al., 2025) | $3.40_{\pm 0.03}$ | $1.52_{\pm 0.05}$ | $4.23_{\pm 0.03}$ | $1213_{\pm 6}$ | $614_{\pm 5}$ | $1172_{\pm 8}$ |
| SMC (Wu et al., 2023) | $3.22_{\pm 0.01}$ | $1.38_{\pm 0.02}$ | $4.09_{\pm 0.01}$ | $1107_{\pm 4}$ | $561_{\pm 3}$ | $1103_{\pm 4}$ |
| SVDD (Li et al., 2024) | $3.14_{\pm 0.04}$ | $1.41_{\pm 0.02}$ | $4.02_{\pm 0.02}$ | $1140_{\pm 5}$ | $557_{\pm 2}$ | $1077_{\pm 6}$ |
| TFG-Flow (Lin et al., 2025) | $2.42_{\pm 0.02}$ | $1.28_{\pm 0.03}$ | $3.12_{\pm 0.01}$ | $902_{\pm 3}$ | $476_{\pm 5}$ | $\underline{981}_{\pm 6}$ |
| **GILC-DB (Ours)** | $\underline{2.23}_{\pm 0.03}$ | $\underline{1.09}_{\pm 0.04}$ | $\underline{2.87}_{\pm 0.04}$ | $\underline{810}_{\pm 8}$ | $\underline{448}_{\pm 4}$ | $988_{\pm 5}$ |
| **GILC-PG (Ours)** | $\mathbf{1.53}_{\pm 0.02}$ | $\mathbf{0.808}_{\pm 0.04}$ | $\mathbf{2.15}_{\pm 0.02}$ | $\mathbf{783}_{\pm 5}$ | $\mathbf{374}_{\pm 7}$ | $\mathbf{875}_{\pm 7}$ |

tasks on natural image datasets.

1) *Class-conditional image generation:* We build upon the discrete diffusion model framework trained on CIFAR-10 (Campbell et al., 2022). The target class labels are utilized as guidance signals to steer the generation process.

2) *Text-to-image generation:* Following recent advancements, we adopt a text-to-image masked generative model termed Meissonic (Bai et al., 2024). To optimize the visual appeal of the generated outputs, we employ the aesthetic score (Pressman et al., 2022) as the reward function, which presents a challenging, non-differentiable optimization objective in high-dimensional pixel spaces.

***Results.*** Both GILC-DB and GILC-PG successfully scale to these complex image domains without requiring any architectural modifications or additional training. For class-conditional generation on CIFAR-10, qualitative evaluations demonstrate that GILC produces distinct, class-consistent images with high sample quality, confirming its efficacy in large-scale discrete state spaces (Fig. 8). Furthermore, in the more challenging text-to-image synthesis task with Meissonic, GILC remains highly effective, successfully generating high-resolution, visually appealing images aligned with the aesthetic score (Fig. 9).

## 5. Discussions and Limitations

This paper introduces the **G**radient-**I**nformed **L**ogit **C**orrection (**GILC**) framework, a versatile framework for guiding discrete diffusion in a plug-and-play manner. By employing a variational proxy to estimate the value function and utilizing gradient-informed logit correction, our method enables precise control over discrete data generation without requiring retraining. Through extensive evaluation, GILC demonstrates robust performance and high adaptability across diverse scientific domains.

While our approach reduces the computational overhead, specifically the number of calls to the diffusion network, it is not without limitations. Both GILC-PG and GILC-DB still necessitate multiple Monte Carlo samples to query the reward function. Further minimizing this sampling requirement while preserving guidance fidelity remains a promising direction for future research. Additionally, our current training-free variational proxy operates under a mean-field assumption of token independence. This assumption can introduce non-trivial errors in structured domains like natural language, where token dependencies are critical. Extending our framework to large-scale diffusion language models by relaxing this independence constraint, as in Xie et al. (2025), represents a significant next step.

## Acknowledgements

This work was supported by the National Natural Science Foundation of China under Grant 62405014 and 62325101.

## Impact Statement

Our proposed framework provides a training-free solution that enables discrete diffusion models to satisfy complex constraints. Owing to its versatility and computational efficiency, we anticipate that this framework will have a positive impact on related research domains, including gene therapy and drug design. Nevertheless, careful consideration is required to prevent potential misuse of the method, underscoring the need for robust ethical oversight.

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

# A. Related Works

We review the landscape of diffusion models for constrained generation, focusing on the evolution from standard guidance to recent developments in diffusion with discrete state spaces.

**Classifier and Classifier-free Guidance.** Classifier Guidance (CG) and Classifier-free Guidance (CFG) are the foundational frameworks for conditional diffusion generation (Dhariwal & Nichol, 2021; Song et al., 2021; Ho & Salimans, 2022). Recent efforts (Nisonoff et al., 2025; Schiff et al., 2025) have extended these paradigms to discrete diffusion and flow models. However, both face significant practical hurdles: CG necessitates training a dedicated noise-aware classifier, which precludes the use of pre-trained models optimized solely for clean data. Conversely, CFG requires paired datasets for joint training, a requirement that limits flexibility and often leads to suboptimal performance in data-scarce regimes.

**Training-free Guidance.** To bypass the need for noise-aware classifiers, several variants have emerged that utilize approximations to estimate the guidance gradient. While these are effective for inverse problems and continuous conditional generation (Chung et al., 2023; Yu et al., 2023; Song et al., 2023a; Bansal et al., 2023; Song et al., 2023b), their application to discrete diffusion is non-trivial due to the discrete nature of the sampling state space. Alternative strategies, such as Sequential Monte Carlo (SMC) (Wu et al., 2023) and SVDD (Li et al., 2024), rely on particle filtering or candidate selection. However, these methods are inherently limited to the explored candidate space; as the sequence length expands, the number of particles required grows rapidly, leading to prohibitive sampling times. Furthermore, while TFG-low (Lin et al., 2025) was designed for multimodal flows, our empirical results suggest it struggles when guiding discrete diffusion in isolation. Other recent techniques involving variable splitting (Chu et al., 2026) show promise but are currently restricted to discrete diffusion with uniform transition matrices. Recently, Ou et al. (2026) suggests improving SMC by using reward gradient guidance as a better proposal distribution. In contrast, we directly derive our value function estimate from the variational objective and directly apply guidance to logits, which we found to be crucial for discrete diffusion.

**Fine-tuning and Reward Optimization.** Conditional generation can also be framed as maximizing a reward function relative to a pre-trained model. While direct backpropagation (Prabhudesai et al., 2023; Clark et al., 2024) and reinforcement learning (Black et al., 2024) have been used to fine-tune continuous models, extensions to discrete states (Venkatraman et al., 2024; Rector-Brooks et al., 2025; Zekri & Boullé, 2026; Wang et al., 2025) are still gaining traction. Unlike these approaches, our method adopts a *plug-and-play* fashion. By utilizing reward functions without parameter updates, we avoid the heavy computational overhead of fine-tuning and mitigate the common pitfall of reward hacking. Tab. 5 shows a comparison between our proposed GILC framework and representative methods.

*Table 5.* Comparison of guidance strategies for discrete diffusion in terms of plug-and-play capability (PnP), inference efficiency (NFE-Eff.), compatibility with non-differentiable objectives (Non-Diff.), and the ability to gradient utilization (Grad. Use). Methods marked ($^\dagger$) require fine-tuning of the diffusion network.

| Method | PnP | NFE-Eff. | Non-Diff. | Grad. Use |
|---|---|---|---|---|
| **GILC (Ours)** | ✓ | ✓ | ✓ | ✓ |
| CG/CFG (Nisonoff et al., 2025) | ✗ | ✓ | ✗ | ✓ |
| SMC (Wu et al., 2023) | ✓ | ✗ | ✓ | ✗ |
| SVDD (Li et al., 2024) | ✓ | ✗ | ✓ | ✗ |
| TFG-Flow (Lin et al., 2025) | ✓ | ✓ | ✓ | ✗ |
| DDPP$^\dagger$ (Rector-Brooks et al., 2025) | ✗ | ✓ | ✓ | ✓ |
| DRAKES$^\dagger$ (Wang et al., 2025) | ✗ | ✓ | ✗ | ✓ |

# B. Additional Method Details

## B.1. Derivation of the Factorized Reverse Process

In Sec. 3.1, we approximate the optimal reverse process $p_\theta^r(\mathbf{z}_s|\mathbf{z}_t)$ using a first-order Taylor expansion to ensure computational tractability.

**Taylor Expansion and Decomposition.** The optimal reverse process is defined as:

$$p_\theta^r(\mathbf{z}_s|\mathbf{z}_t) = \frac{p_\theta(\mathbf{z}_s|\mathbf{z}_t)\exp(v(\mathbf{z}_s)/\beta)}{\sum_{\mathbf{z}_s} p_\theta(\mathbf{z}_s|\mathbf{z}_t)\exp(v(\mathbf{z}_s)/\beta)}$$

Applying a first-order Taylor expansion to $v(\mathbf{z}_s)$ around $\mathbf{z}_t$:

$$v(\mathbf{z}_s) \approx v(\mathbf{z}_t) + \sum_{\ell=1}^{L} \langle \mathbf{z}_s^{\ell} - \mathbf{z}_t^{\ell}, \nabla_{\mathbf{z}_t^{\ell}} v(\mathbf{z}_t) \rangle$$

Substituting this into the exponential term and noting that $\exp(v(\mathbf{z}_t)/\beta)$ is a constant that cancels out in the normalization, we obtain:

$$\exp(v(\mathbf{z}_s)/\beta) \propto \prod_{\ell=1}^{L} \exp\left( \frac{1}{\beta} \langle \mathbf{z}_s^{\ell} - \mathbf{z}_t^{\ell}, \nabla_{\mathbf{z}_t^{\ell}} v(\mathbf{z}_t) \rangle \right)$$

**Factorization of the Partition Function.** Assuming a factorized base transition $p_\theta(\mathbf{z}_s | \mathbf{z}_t) = \prod_{\ell=1}^{L} p_\theta(\mathbf{z}_s^{\ell} | \mathbf{z}_t)$, the optimal process becomes:

$$p_\theta^r(\mathbf{z}_s | \mathbf{z}_t) \approx \frac{\prod_{\ell=1}^{L} p_\theta(\mathbf{z}_s^{\ell} | \mathbf{z}_t) \exp\left( \frac{1}{\beta} \langle \mathbf{z}_s^{\ell} - \mathbf{z}_t^{\ell}, \nabla_{\mathbf{z}_t^{\ell}} v(\mathbf{z}_t) \rangle \right)}{\sum_{\mathbf{z}_s} \prod_{\ell=1}^{L} p_\theta(\mathbf{z}_s^{\ell} | \mathbf{z}_t) \exp\left( \frac{1}{\beta} \langle \mathbf{z}_s^{\ell} - \mathbf{z}_t^{\ell}, \nabla_{\mathbf{z}_t^{\ell}} v(\mathbf{z}_t) \rangle \right)}$$

Using the identity $\sum_{\mathbf{z}_s} \prod_{\ell=1}^{L} f_\ell(\mathbf{z}_s^{\ell}) = \prod_{\ell=1}^{L} \sum_{\mathbf{z}_s^{\ell}} f_\ell(\mathbf{z}_s^{\ell})$, the global normalization constant (the partition function) factorizes into $L$ independent local sums:

$$p_\theta^r(\mathbf{z}_s | \mathbf{z}_t) \approx \prod_{\ell=1}^{L} \frac{p_\theta(\mathbf{z}_s^{\ell} | \mathbf{z}_t) \exp\left( \frac{1}{\beta} \langle \mathbf{z}_s^{\ell} - \mathbf{z}_t^{\ell}, \nabla_{\mathbf{z}_t^{\ell}} v(\mathbf{z}_t) \rangle \right)}{\sum_{k=1}^{K} p_\theta(\mathbf{z}_s^{\ell} = k | \mathbf{z}_t) \exp\left( \frac{1}{\beta} \langle k - \mathbf{z}_t^{\ell}, \nabla_{\mathbf{z}_t^{\ell}} v(\mathbf{z}_t) \rangle \right)} = \prod_{\ell=1}^{L} p_\theta^r(\mathbf{z}_s^{\ell} | \mathbf{z}_t)$$

As a result, the final probability is fully decomposable across sequence positions. This allows each dimension $\ell$ to perform sampling independently, transforming an exponentially complex joint distribution into $L$ parallelizable categorical distributions.

## B.2. Formal Derivation of Logit Guidance

In this section, we demonstrate that for discrete diffusion models utilizing an absorbing state, the optimal reward-guided reverse process of GILC is exactly equivalent to applying a correction to the model's prediction logits.

**Proof of Equivalence:** $p_\theta^r(\mathbf{z}_s \mid \mathbf{z}_t) = q(\mathbf{z}_s \mid \mathbf{z}_t, \mathbf{x}_\theta^r)$. In an absorbing state discrete diffusion model, let $m$ denote the $[\text{MASK}]$ state. When the current latent $\mathbf{z}_t$ is masked, the base reverse transition for a single token is given by the categorical distribution:

$$p_\theta(\mathbf{z}_s = k \mid \mathbf{z}_t = m) \propto \begin{cases} (\alpha_s - \alpha_t)\mathbf{x}_{\theta,k} & k \neq m \\ 1 - \alpha_s & k = m \end{cases}$$

where $\mathbf{x}_{\theta,k} = \frac{\exp(\eta_k)}{\sum_j \exp(\eta_j)}$ represents the model's predicted probability for the clean-data class $k$.

The optimal guided distribution $p_\theta^r$ (Eq. 6) is defined by the reward-tilted density:

$$p_\theta^r(\mathbf{z}_s = k \mid \mathbf{z}_t) \propto p_\theta(\mathbf{z}_s = k \mid \mathbf{z}_t) \exp\left( \frac{v(\mathbf{z}_s = \mathbf{e}_k)}{\beta} \right)$$

where $\mathbf{e}_k$ is the one-hot encoding for class $k$. We approximate the value function $v(\mathbf{z}_s = \mathbf{e}_k)$ via a first-order expansion around $\mathbf{z}_t$. Using the Jacobian-free correction vector $g_\eta$ to represent the gradient direction, and omitting terms independent of $k$ (which vanish under the proportionality), the value term is approximated as:

$$\exp\left( \frac{v(\mathbf{z}_s = \mathbf{e}_k)}{\beta} \right) \approx \exp\left( \frac{v(\mathbf{z}_t) + \langle g_\eta, \mathbf{e}_k - \mathbf{z}_t \rangle}{\beta} \right) \propto \exp\left( \frac{g_{\eta,k}}{\beta} \right)$$

where $g_{\eta,k} = \langle \mathbf{e}_k, g_\eta \rangle$ selects the $k$-th component of the correction vector $g_\eta$. Substituting the Softmax form of $\mathbf{x}_{\theta,k}$ into the above equation for $k \neq m$:

$$p_\theta^r(\mathbf{z}_s = k \mid \mathbf{z}_t) \propto \mathbf{x}_{\theta,k} \cdot \exp\left(\frac{g_{\eta,k}}{\beta}\right) = \left(\frac{\exp(\eta_k)}{\sum_j \exp(\eta_j)}\right) \exp\left(\frac{g_{\eta,k}}{\beta}\right)$$

$$\propto \exp\left(\eta_k + \frac{g_{\eta,k}}{\beta}\right)$$

Normalizing this categorical distribution over the vocabulary $j \in \{1, \ldots, K\}$ yields:

$$p_\theta^r(\mathbf{z}_s = k \mid \mathbf{z}_t) = \frac{\exp(\eta_k + g_{\eta,k}/\beta)}{\sum_j \exp(\eta_j + g_{\eta,j}/\beta)}$$

Conversely, we define the guided prediction as $\mathbf{x}_\theta^r = \mathrm{Softmax}(\eta + g_\eta/\beta)$. Substituting this into the standard posterior $q$ (Eq. 3) for the unmasking step:

$$q(\mathbf{z}_s = k \mid \mathbf{z}_t, \mathbf{x}_\theta^r) = \frac{(\alpha_s - \alpha_t)\mathbf{x}_{\theta,k}^r + (1 - \alpha_s)\mathbb{I}[k = m]}{1 - \alpha_t}$$

For any non-mask token $k \neq m$, the term matches the result in $p_\theta^r(\mathbf{z}_s = k \mid \mathbf{z}_t)$ exactly (under the transition kernel's normalization). Thus, the reward-tilted process is equivalent to a standard unmasking step using corrected clean prediction:

$$p_\theta^r(\mathbf{z}_s \mid \mathbf{z}_t) = q(\mathbf{z}_s \mid \mathbf{z}_t, \mathbf{x}_\theta^r)$$

This completes the proof.

### B.3. Pseudocodes

This section presents the pseudocode implementation of the GILC framework through Algorithm 1. Algorithms 2 and 3 respectively detail the gradient computation for GILC-DB and CILC-PG.

---

**Algorithm 1** Gradient-Informed Logit Correction (GILC)

---

1: **Input:** Pre-trained discrete diffusion model $p_\theta$, reward function $r(\cdot)$, Monte Carlo number $n$, guidance scale $\beta^{-1}$
2: Initialize fully masked sequence $\mathbf{z}_T \leftarrow \mathbf{m}$ ;
3: **for** $t = T, \ldots, 1$ **do**
4:     Predict clean data logits $s \leftarrow t - 1$, $\eta \leftarrow p_\theta(\mathbf{z}_s | \mathbf{z}_t)$ ;
5:     Estimate correction gradient $g_\eta \leftarrow$ Correction-DB/PG$(\eta, r, n)$ (Algorithms 2 or 3) ;
6:     Apply logit guidance $\eta^r \leftarrow \eta + g_\eta/\beta$ ;
7:     Compute guided prediction $\mathbf{x}_\theta^r \leftarrow \mathrm{Softmax}(\eta^r)$ ;
8:     Sample the next state $\mathbf{z}_s \sim q(\mathbf{z}_s \mid \mathbf{z}_t, \mathbf{x}_\theta^r)$ (Eq. 3) ;
9: **end for**
10: **Output:** Generated sample $\mathbf{x} \leftarrow \mathbf{z}_0$

---

**Algorithm 2** Correction-DB (Direct Backpropagation Estimator)

---

1: **Input:** Logits $\eta$, reward function $r(\cdot)$, sample size $n$
2: Sample soft samples $\mathbf{x}_{\mathrm{soft}}^{(1)}, \cdots, \mathbf{x}_{\mathrm{soft}}^{(n)} \sim \mathrm{Gumbel\text{-}Softmax}(\eta)$ ;
3: Compute straight-through samples $\hat{\mathbf{x}}^{(i)} \leftarrow \mathrm{onehot}\left(\arg\max \mathbf{x}_{\mathrm{soft}}^{(i)}\right) - \mathrm{sg}\left(\mathbf{x}_{\mathrm{soft}}^{(i)}\right) + \mathbf{x}_{\mathrm{soft}}^{(i)}$, $i = 1, \ldots, n$ ;
4: Evaluate rewards $R_i \leftarrow r(\hat{\mathbf{x}}^{(i)})$, $i = 1, \cdots, n$ ;
5: Estimate gradient with backpropgation $g_\eta \leftarrow \frac{1}{n} \sum_{i=1}^n \frac{\partial r(\hat{\mathbf{x}}^{(i)})}{\partial \hat{\mathbf{x}}^{(i)}} \frac{\partial \hat{\mathbf{x}}^{(i)}}{\partial \eta}$ ;
6: **Output:** Logit correction $g_\eta$

---

---

**Algorithm 3** Correction-PG (Policy Gradient Estimator)

---

1: **Input:** Logits $\eta$, reward function $r(\cdot)$, sample size $n$
2: Sample a group of candidates $\mathbf{x}^{(1)}, \cdots, \mathbf{x}^{(n)} \sim \text{Cat}(\mathbf{x}; \mathbf{x}_\theta(\mathbf{z}_t, t) = \text{Softmax}(\eta))$ ;
3: Evaluate rewards $R_i \leftarrow r(\mathbf{x}^{(i)}), \ i = 1, \cdots, n$ ;
4: Compute group relative advantages $\mathcal{A}_i \leftarrow \frac{R_i - \text{mean}(\{R_1, \cdots, R_n\})}{\text{std}(\{R_1, \cdots, R_n\}}$ ;
5: Estimate policy gradient: $g'_\eta \leftarrow \frac{1}{n} \sum_{i=1}^{n} \mathcal{A}_i \frac{\partial \log \langle \mathbf{x}_\theta(\mathbf{z}_t, t), \mathbf{x}^{(i)} \rangle}{\partial \eta}$ ;
6: **Output:** Logit correction $g'_\eta$

---

## C. Experimental Details

This section describes the implementation details and hyperparameter settings of all baseline methods. Quantitative results for classifier-based and classifier-free guidance, as well as fine-tuning methods, are primarily drawn from existing literature. For training-free baselines, we faithfully reproduced the original experimental setups to ensure fair and rigorous comparisons with our approach. All experiments were conducted on a single NVIDIA A100 GPU with 80 GB of memory.

### C.1. Implementation Details of Methods

In this section, we describe the implementation details of the baseline methods used for comparison.

**Classifier Guidance (CG)** (Nisonoff et al., 2025). Classifier Guidance (CG) steers generation toward target attributes during sampling by leveraging an auxiliary noise-aware classifier. For comparison, we implement CG following the procedures described in Wang et al. (2025) and Li et al. (2024). Specifically, the predictor is estimated using posterior mean estimation (Chung et al., 2023). This involves first extracting a denoised sequence from the noisy input $\mathbf{z}_t$ using a pretrained diffusion model, and then evaluating the predicted clean sequence with the reward oracle. However, unlike in continuous diffusion settings, we observe that this posterior mean–based estimation is ineffective in discrete diffusion models. As a result, CG fails to provide reliable guidance signals in our setting, which explains its relatively poor performance reported in the experimental results.

**Classifier-free Guidance (CFG)** (Nisonoff et al., 2025). In contrast to CG, Classifier-Free Guidance (CFG) trains conditional generation models from scratch (Ho & Salimans, 2022). We adopt the CFG implementation provided by Wang et al. (2025). To generate sequences with desired properties, CFG incorporates reward values as additional conditioning inputs to the diffusion model and encourages sampling toward high-reward regions. Concretely, reward values are binarized using the 95th percentile as the threshold, and sampling is performed conditioned on the high-reward label. We emphasize that CFG requires access to labeled training pairs with associated reward values. Consequently, its performance may degrade in regimes where labeled data are scarce.

**DRAKES** (Wang et al., 2025). DRAKES is a method for directly fine-tuning discrete diffusion models via reward optimization. It enables backpropagation through the reward signal during sampling by maintaining differentiability using reparameterization techniques. In practice, DRAKES employs truncation strategies and observes that initiating backpropagation from intermediate diffusion time steps is often more effective than propagating gradients from the initial noise state. To further stabilize training and mitigate reward hacking, DRAKES introduces a KL-divergence regularization term that constrains the fine-tuned model to remain close to the pretrained distribution. In our experiments, we adopt the fine-tuned models released by Wang et al. (2025) and report their quantitative performance.

**Best-of-$N$** (Beirami et al., 2025). Best-of-$N$ is a simple yet effective baseline for guiding generative models, wherein multiple candidate samples are generated independently and the one with the highest reward is selected. In our experiments, we set the number of samples to $N = 20$.

**Sequential Monte Carlo (SMC)** (Wu et al., 2023). SMC is a general-purpose sampling framework that maintains a population of particles and applies filtering operations during generation to approximate the target distribution. While SMC is theoretically exact in the limit of infinitely many particles, practical implementations necessarily operate with a finite particle budget. In our experiments, we set the number of particles to $N = 20$.

**SVDD** (Li et al., 2024). SVDD is a derivative-free guidance method for diffusion models. At each diffusion time step, it samples multiple candidate states from the transition kernel, estimates the future value of each candidate using a deterministic

evaluator, and selects the candidate with the highest estimated value as the next state. In our experiments, we set the number of candidates to $N = 20$.

**TFG-Flow** (Lin et al., 2025). TFG-Flow was originally proposed as a training-free guided multimodal flow model that jointly generates continuous and discrete components. When restricted to discrete guidance, it can also be applied as a discrete diffusion model. TFG-Flow requires estimating a guided rate matrix; in our experiments, this estimation is performed using $N = 20$ samples. All other hyperparameters are kept consistent with the official implementation.

**GILC-DB/PG (Ours)**. Our framework includes several tunable hyperparameters for guidance optimization. For GILC-DB, we use 20 Monte Carlo samples for value estimation in the protein unfolding task and 5 samples for DNA and small molecule generation, which we find sufficient to achieve state-of-the-art performance. The Gumbel-Softmax temperature $\tau$ is fixed at 1.0. For GILC-PG, the number of samples is set to 20 across all tasks. The guidance strength $\beta$ is selected via grid search, yielding values of 10,000 for DNA, 1,000 for protein sequences, and 5,000 for small molecules.

## C.2. Use of Models

All experiments were conducted using pre-trained models, with parameters kept frozen during guided sampling. For the DNA and protein sequence tasks, we employed a pre-trained discrete diffusion model trained on enhancer sequences and a pre-trained reverse folding model, respectively. The model architectures and associated reward functions were adopted directly from the repository provided by Wang et al. (2025). For the molecular generation task, we utilized a pre-trained multimodal flow model based on the EGNN architecture, as implemented by Lin et al. (2025). All reward functions and attribute predictors remained fixed throughout the experiments and were accessed solely during the inference phase.

## C.3. Evaluation metrics

The following is a detailed introduction to the metrics used in our experiment.

**DNA Sequence Design.** Following established conventions (Wang et al., 2025), we evaluate the generated enhancer sequences using the following metrics:

- **Predicted Activity (Pred-Activity):** We measure the enhancer activity level in the HepG2 cell line using a reward oracle trained on a held-out evaluation subset. Crucially, the oracle used for guidance (or fine-tuning) is trained on a disjoint subset of data split by chromosomes, ensuring zero overlap with the evaluation oracle.

- **Chromatin Accessibility (ATAC-Acc):** To validate whether the synthetic sequences correspond to accessible chromatin regions (a hallmark of active enhancers), we utilize an independent binary classification model trained on HepG2 chromatin accessibility data. While this metric is not used for optimization, it serves as an external validation of sequence biological plausibility.

- **3-mer Pearson Correlation (3-mer Corr):** We assess the distributional similarity between generated sequences and high-activity natural sequences. We calculate the Pearson correlation of 3-mer counts between the synthetic samples and the top 0.1% of sequences with the highest HepG2 activity in the reference dataset.

- **JASPAR Motif Analysis (JASPAR Corr):** We analyze the biological relevance of generated motifs using JASPAR transcription factor binding profiles. We calculate the Spearman correlation of motif frequencies between the generated sequences and the top 0.1% of natural high-activity sequences, verifying if the model captures the motif patterns driving enhancer activity.

- **Approximated Log-Likelihood (App-Log-Lik):** To quantify how natural the generated sequences appear to the pre-trained base model, we compute the approximate log-likelihood using the Evidence Lower Bound (ELBO) of the discrete diffusion model. Lower likelihoods indicate out-of-distribution sequences that may result from over-optimizing the reward oracle.

**Protein Inverse Folding.** We evaluate the stability of generated sequences and their structural consistency with the target backbone. Note that for all evaluations, we condition on protein backbone conformations from the test set that were unseen during fine-tuning.

- **Predicted Stability (Pred-ddG):** We use an evaluation oracle trained on the full Megascale dataset (train, validation, and test splits) to predict protein stability ($\Delta\Delta G$). In contrast, the oracle used for guidance during sampling is trained strictly on the Megascale training set, preventing information leakage.

- **Self-consistency RMSD (scRMSD):** To verify if the generated sequence folds into the target structure, we predict the structure of the generated sequence using ESMFold (**?**) and calculate the Root Mean Square Deviation (RMSD) between the predicted structure and the ground-truth wild-type backbone.

- **Success Rate:** Following Campbell et al. (2024), we also define the success rate as the proportion of generated sequences that satisfy both stability and structural constraints: specifically, Pred-ddG > 0 and scRMSD < 2 Å.

**Molecular Generation.**

- **Mean Absolute Error (MAE):** To evaluate the alignment between generated molecules and the target property, we follow the protocol from (Bao et al., 2023). The MAE is calculated as:

$$\text{MAE} = \frac{1}{M} \sum_{i=1}^{M} |\phi_p(\mathbf{x}_i) - c_i|$$

where $\phi_p$ is the evaluation predictor, $\mathbf{x}_i$ represents a generated molecule and $c_i$ denotes its target property.

## D. Additional Results

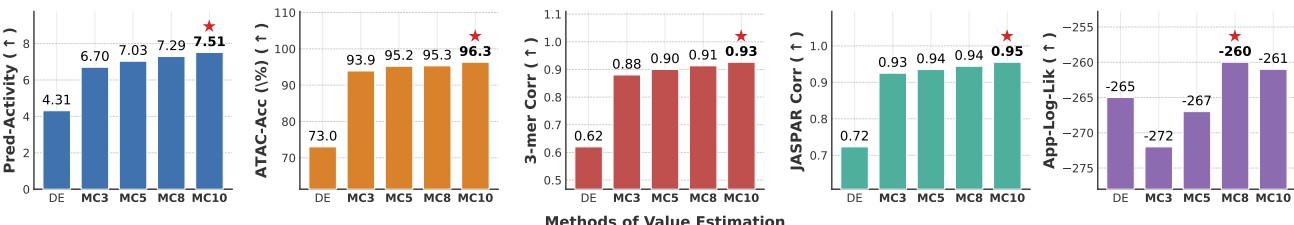

*Figure 5.* Performance comparison on DNA sequence design. The deterministic estimation (DE) baseline is compared with Monte Carlo (MC) estimation using varying sample sizes ($n = 3 \sim 10$). Metrics include predicted activity, chromatin accessibility (ATAC-Acc), motif correlations (3-mer, JASPAR), and log-likelihood. Red stars ($\star$) indicate the best performance for each metric.

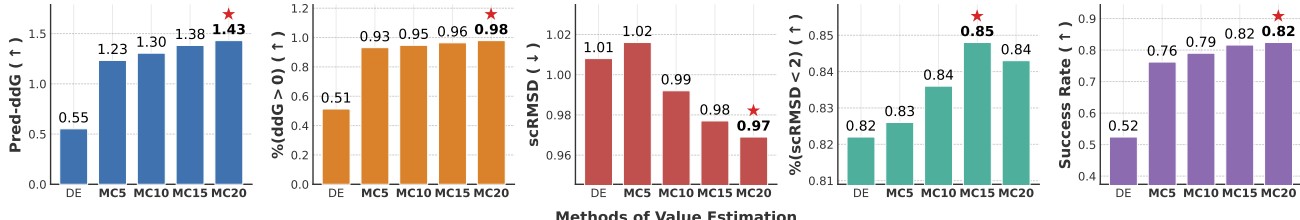

*Figure 6.* Performance comparison on protein sequence design. The deterministic estimation (DE) baseline is compared with Monte Carlo (MC) estimation using varying sample sizes ($n = 5 \sim 20$). Metrics evaluate stability (Pred-ddG, positive proportion) and structural self-consistency (scRMSD) and success rate. Red stars ($\star$) indicate the best performance for each metric.

### D.1. Result of Molecular Generation

In Tab. 4, the *Upper bound* corresponds to a diagnostic baseline obtained by randomly shuffling attribute labels within the unseen half of the QM9 training set, thereby removing correlations between molecular structures and target properties. The MAE is then computed using this shuffled dataset. If a method performs better than this upper bound, it indicates that the model successfully incorporates conditional attribute information into the generated molecules rather than relying on spurious correlations. The *#Atoms* baseline predicts molecular properties solely based on the number of atoms in the molecule. Performance surpassing this baseline suggests that the model captures conditional information beyond simple

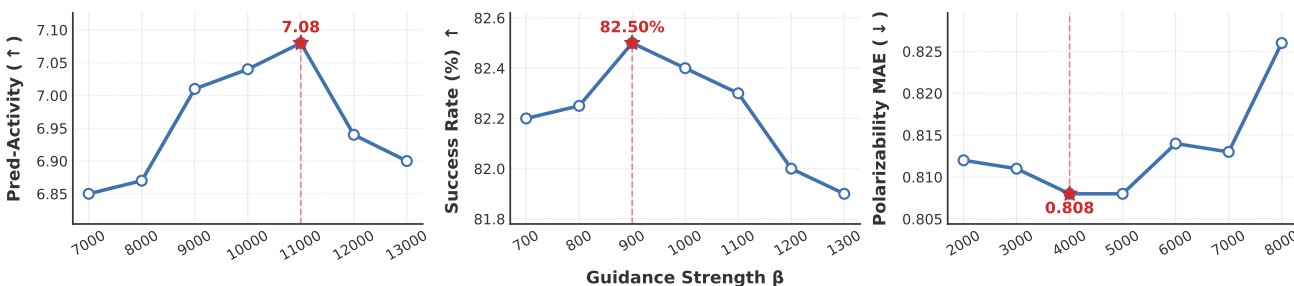

*Figure 7.* Ablation study on guidance strength $\beta$ on GILC-DB. The impact of varying guidance strength is evaluated across three tasks. Red stars ($\star$) indicate the optimal performance for each metric.

size-related cues in the generated molecular structures. Finally, the *Lower bound* represents the MAE achieved by directly predicting target properties using the pretrained property predictor itself, serving as an approximate oracle reference (Bao et al., 2023).

To evaluate framework generality under more challenging scenarios, we incorporate a structure-guided molecular generation task constrained by non-differentiable reward functions. Here, molecular structures are characterized using discrete molecular fingerprints (Gebauer et al., 2022), and the Tanimoto coefficient (Bajusz et al., 2015) is utilized to measure structural similarity against target molecules. As reported in Tab. 6, GILC-PG significantly outperforms all baselines, achieving the highest similarity score. Overall, these findings highlight the effectiveness and generality of the GILC framework in guiding discrete and multimodal flows toward both differentiable and non-differentiable targets without requiring additional training.

*Table 6.* Structural similarity results on the QM9 dataset under non-differentiable fingerprint rewards.

| Method | Similarity ↑ |
|---|---|
| Best-of-$N$ (Beirami et al., 2025) | $0.182_{\pm 0.016}$ |
| SMC (Wu et al., 2023) | $0.178_{\pm 0.002}$ |
| SVDD (Li et al., 2024) | $0.234_{\pm 0.011}$ |
| TFG-Flow (Lin et al., 2025) | $0.271_{\pm 0.006}$ |
| **GILC-PG (Ours)** | $\mathbf{0.308}_{\pm 0.004}$ |

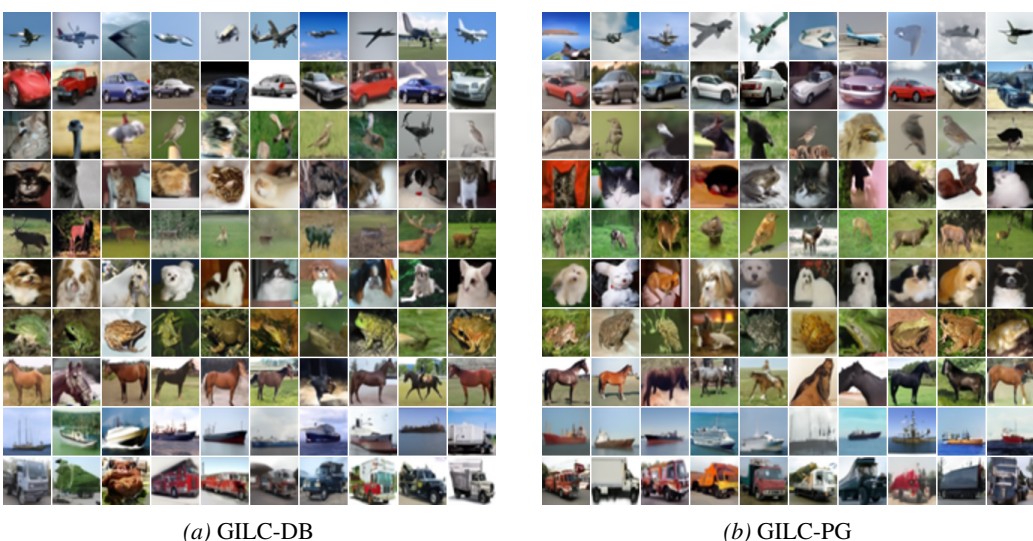

*(a)* GILC-DB          *(b)* GILC-PG

*Figure 8.* Class-conditional images generated by GILC-DB and GILC-PG on CIFAR-10.

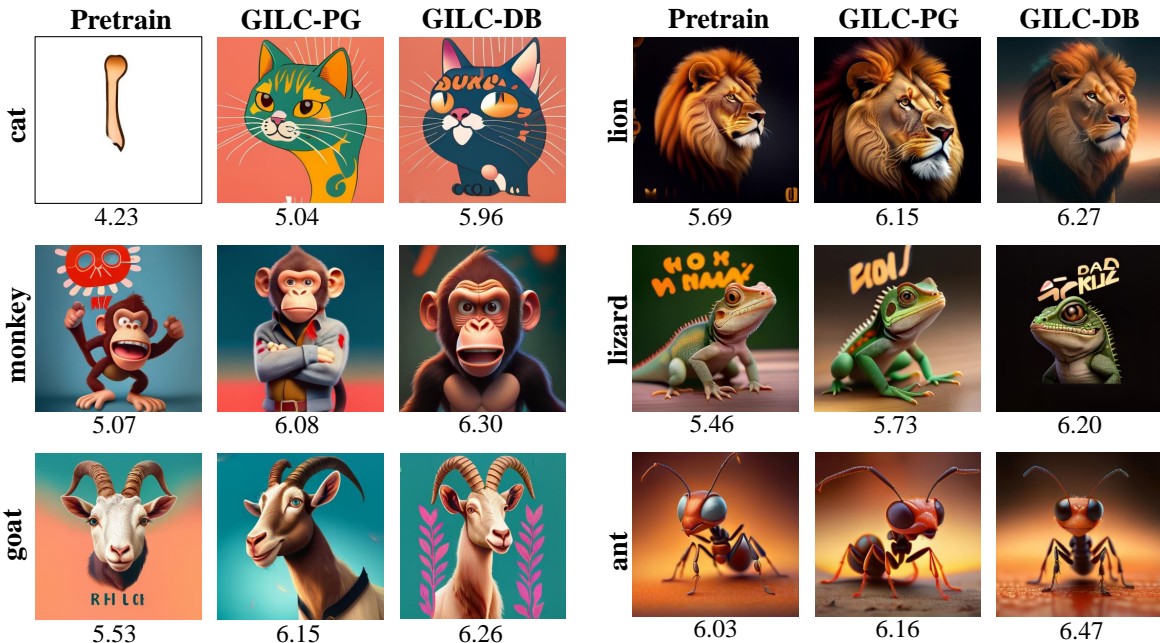

*Figure 9.* Improving aesthetic score for text-to-image generation. The aesthetic score is shown at the bottom of the image.

## D.2. Result of Image Discrete Diffusion

Figs. 8 and 9 present qualitative visualizations to intuitively demonstrate the generation quality and guidance efficacy of our proposed framework on two distinct image discrete diffusion tasks: class-conditional generation on CIFAR-10 (Campbell et al., 2022) and high-resolution text-to-image synthesis via Meissonic (Bai et al., 2024).

## D.3. Ablation Study

We first perform a series of ablation studies to rigorously validate the effectiveness of our framework's core components. Subsequently, we investigate the sensitivity of GILC-DB and GILC-PG to various hyperparameter configurations.

**Value Function Estimation.** To validate the efficacy of our proposed variational proxy method for value function estimation, we conduct an ablation study by substituting this component with deterministic estimation methods from the literature (Li et al., 2024) while holding all other experimental factors constant. Evaluations are performed on DNA sequence design and protein inverse folding tasks. As illustrated in Figs. 5 and 6, our variational proxy consistently yields more effective guidance than deterministic estimators. Notably, increasing the number of Monte Carlo samples further enhances estimation accuracy, directly translating to superior performance. This suggests that the variational approach captures the underlying distribution of the discrete state space more effectively than point-based estimates.

**Impact of the Model Jacobian.** We further evaluate the effect of excluding the discrete diffusion model Jacobian. Specifically, we compare generation results when the corrected gradient is computed with respect to both the clean prediction logits and the noisy state, versus when the model Jacobian term is omitted. As shown in Tabs. 7 and 8, removing the Jacobian term consistently improves guidance performance across tasks. This empirical observation is consistent with our theoretical analysis. In high-dimensional discrete settings, the model Jacobian is often severely ill-conditioned, which can introduce numerical instability into gradient-based guidance. By excluding this term, we obtain a more stable and reliable guidance signal, enabling more robust convergence toward the desired target attributes.

**Effect of Guidance Strength $\beta$.** Our method relies on gradient-based corrections, making the guidance strength $\beta$ a critical hyperparameter. As derived in Eq. 6, $\beta$ controls the magnitude of the guidance signal. We visualize its effect on DNA enhancer optimization, protein stability enhancement, and molecular polarizability $\alpha$ guidance. As shown in Fig. 7, an appropriately chosen $\beta$ leads to strong performance. In contrast, overly large values of $\beta$ can reduce sample diversity and hinder exploration during early sampling, while overly small values provide insufficient guidance, both resulting in degraded performance.

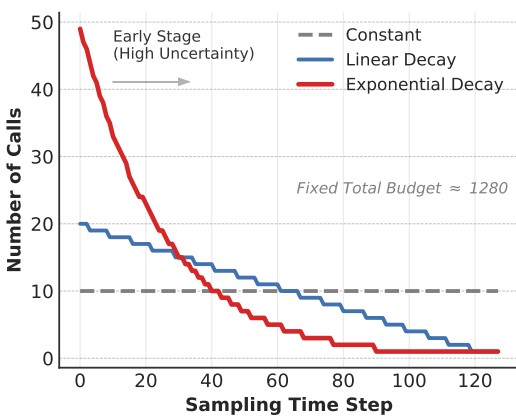

*Figure 10.* Illustration of constant, linear, and exponential decay strategies normalized to a fixed budget ($N \approx 128 \times 10$). The exponential decay schedule prioritizes early-stage evaluations to counter high initial uncertainty.

**Allocation of Reward Function Calls.** As shown in Fig. 2a, estimation errors of the value function are typically larger during the early stages of sampling. This suggests that, under a fixed budget of reward function evaluations, allocating more calls to early time steps is more effective than distributing them uniformly. We evaluate several scheduling strategies, including constant, linear decay, and exponential decay schedules (shown in Fig. 10). Our results in Tabs. 9 and 10 show that both linear and exponential decay schedules, which emphasize early-stage evaluations, substantially improve performance. Accordingly, we adopt exponential decay scheduling as the default in all experiments.

*Table 7.* Ablation study on logit correction by omitting model Jacobian on regulatory DNA sequence design. **Bold** indicates the best performance within the same method.

| Method | | Pred-Activity ↑ | ATAC-Acc ↑ (%) | 3-mer Corr ↑ | JASPAR Corr ↑ | App-Log-Lik ↑ |
|---|---|---|---|---|---|---|
| **GILC-DB** | *w/* Jacobian | 4.18 | 48.8 | 0.816 | 0.904 | **−263** |
| | *w/o* Jacobian | **7.04** | **95.2** | **0.900** | **0.935** | −267 |
| **GILC-PG** | *w/* Jacobian | 3.44 | 34.7 | 0.518 | 0.673 | **−259** |
| | *w/o* Jacobian | **5.21** | **84.0** | **0.910** | **0.937** | −270 |

*Table 8.* Ablation study on logit correction by omitting model Jacobian on protein sequence design. **Bold** indicates the best performance within the same method.

| Method | | Pred-ddG ↑ | %(ddG> 0) (%)↑ | scRMSD ↓ | %(scRMSD< 2)(%)↑ | Success Rate (%)↑ |
|---|---|---|---|---|---|---|
| **GILC-DB** | *w/* Jacobian | 0.809 | 79.5 | **0.928** | **87.6** | 70.1 |
| | *w/o* Jacobian | **1.430** | **97.9** | 0.968 | 84.3 | **82.4** |
| **GILC-PG** | *w/* Jacobian | 0.528 | 44.7 | 0.941 | 89.2 | 48.9 |
| | *w/o* Jacobian | **0.719** | **75.6** | **0.914** | **92.4** | **69.8** |

*Table 9.* Ablation study of reward call schedules on DNA sequence generation using GILC-DB. The best performance is highlighted in **bold**, and the second best is underlined.

| Schedule | Pred-Activity ↑ | ATAC-Acc ↑ (%) | 3-mer Corr ↑ | JASPAR Corr ↑ | App-Log-Lik ↑ |
|---|---|---|---|---|---|
| Constant | 6.20 | 87.3 | 0.806 | 0.890 | -274 |
| Linear Decay | 6.62 | 95.0 | 0.853 | 0.912 | -270 |
| Exponential Decay | **7.04** | **95.2** | **0.900** | **0.935** | **-267** |

*Table 10.* Ablation study of reward call schedules on protein backbone design using GILC-DB. The best performance is highlighted in **bold**, and the second best is underlined.

| Method | Pred-ddG ↑ | %(ddG> 0) (%) ↑ | scRMSD ↓ | %(scRMSD< 2) (%) ↑ | Success Rate (%) ↑ |
|---|---|---|---|---|---|
| Constant | 1.33 | 93.2 | **0.967** | 83.0 | 76.9 |
| Linear Decay | 1.35 | 95.8 | 1.000 | **85.4** | 81.6 |
| Exponential Decay | **1.43** | **97.9** | 0.968 | 84.3 | **82.4** |

