# OpenReview forum: "Plug-and-Play Guidance for Discrete Diffusion Models via Gradient-Informed Logit Correction"
_ICML.cc/2026/Conference — ICML 2026 regular_

### Official Review · Reviewer_LdoM · 2026-02-15

**Soundness:** 3
**Presentation:** 3
**Significance:** 2
**Originality:** 3
**Overall Recommendation:** 4
**Confidence:** 4

**Summary:**

The paper presents Gradient-Informed Logit Correction (GILC), a plug-and-play framework that estimates guidance signals for discrete diffusion models. The authors introduce variants for both differentiable (-DB) and non-differentiable rewards (-PG). The method employs a variational approach where the pre-trained model serves as a proxy for reward value estimation. It utilizes the Gumbel-Softmax trick to maintain gradient flow in discrete space and stabilizes gradient guidance by discarding the noisy Jacobian term. Comprehensive experiments are conducted to demonstrate the method's effectiveness.

**Compliance With Llm Reviewing Policy:**

Affirmed.

**Final Justification:**

The authors have addressed my concerns, including additional discussions, and empirical analyses. Overall, the responses are satisfactory, and I will keep my score at 4.

**Key Questions For Authors:**

1. While I understand that using the mathematically correct guidance term (Eq. 13) might be unstable and that the simplification in Eq. 14 is well-motivated empirically, do the authors have any theoretical guarantees for this simplification?
2. Can the authors perform a comparative study (e.g. on magnitudes, directions) between the guidance terms derived by GILC-DB and GILC-PG, respectively, in a reward-differentiable setting?
3. Can the authors provide an ablation study examining the effect of changing the GRPO component when applying GILC-PG? How does substituting this with other variance reduction techniques affect performance?

**Limitations:**

1. As the authors mentioned, their current training-free variational proxy operates under a mean-field assumption of token independence, factorizing the joint distribution.
2. The algorithms proposed by the authors lack theoretical guarantees, although this is a relatively minor point given the empirical results.

**Strengths And Weaknesses:**

### Strengths
1. The paper is well-written and clearly introduces the method's motivation and the current problems in the field. The methodology section is easy to follow; the rationale for introducing specific tools to solve problems or enhance performance (such as using the Gumbel-Softmax trick to enable gradient flow through sampling and discarding unstable terms in the estimator) is very clear and well-supported by graphs and experimental figures.
2. A comprehensive experimental study is provided, demonstrating the method's performance.

### Weaknesses
1. There is a mathematical inaccuracy around line 155. The authors state that when $\beta$ is small, the soft value function is approximated by the expected reward. However, mathematically, the expected reward approximation holds when $\beta$ is *large*. When $\beta$ is small, the soft value approaches the maximum reward. Specifically:
   $$
   \lim_{\beta \rightarrow 0}\beta \log \mathbb{E}[\exp(r(x)/\beta)] = \max r(x),\quad \lim_{\beta \rightarrow \infty}\beta \log \mathbb{E}[\exp(r(x)/\beta)] = \mathbb{E}[r(x)].
   $$
   Additionally, there is a minor omission around line 266: when applying the Gumbel-Softmax trick, the Gumbel noise is required to be i.i.d., but the authors do not explicitly mention this. I suggest the authors be more precise with mathematical details.

2. The authors should discuss more recent training-free steering/guidance methods in continuous diffusion, such as [1] and [2]. These works offer different perspectives and represent a fast-evolving area that is relevant to the proposed method.

3. In Table 1, although GILC-DB and GILC-PG achieve decent reward performance, the samples guided by them exhibit relatively lower approximated log-likelihood measures compared to other methods. This suggests potential over-optimization, which could lead to reward hacking.

**References:**

[1] Singhal, Raghav, Zachary Horvitz, Ryan Teehan, Mengye Ren, Zhou Yu, Kathleen McKeown, and Rajesh Ranganath. "A general framework for inference-time scaling and steering of diffusion models." arXiv preprint arXiv:2501.06848 (2025).

[2] Zhang, Xiangcheng, Haowei Lin, Haotian Ye, James Zou, Jianzhu Ma, Yitao Liang, and Yilun Du. "Inference-time scaling of diffusion models through classical search." arXiv preprint arXiv:2505.23614 (2025).

---

> ### Author Rebuttal · Authors · 2026-03-31
>
> Thanks for your constructive feedback and valuable suggestions. Below, we address each comment in detail.
>
> ## Q1. Mathematical inaccuracy around line 155 and Gumbel noise.
>
> Thank you for pointing this out. We adopted the definition of the soft value function from [3], and the statement in the manuscript was indeed a typographical error. The approximation holds when $\beta$ is large, not small, and we will correct this in the revised version. Additionally, we will clarify that the Gumbel noise must be i.i.d.
>
> ## Q2. Discussion on [1] and [2].
>
> [1] proposes a Feynman-Kac steering framework for steering diffusion in continuous models, while [2] uses tree search to enhance inference-time guidance.
>
> Our work shares the same goal with [1,2], maximizing the reward while preserving the baseline model’s distribution (Eq. 5), but differs in focus: we target discrete data generation, whereas these works primarily address continuous generative models, particularly in the image domain. We will incorporate a discussion of these methods in the revised manuscript.
>
>
> ## Q3. Potential over-optimization and reward hacking.
>
> In fact, there is a natural trade-off between maximizing rewards and preserving the unconditional distribution. This trade-off is evident in both DNA generation (Pred-Activity vs. App-Log-Lik) and protein generation (Pred-ddG vs. scRMSD).
>
>  **As a plug-and-play framework, GILC allows users to flexibly adjust the guidance scale at inference time to find the optimal balance for their specific downstream task without retraining.**
>
> GILC also pushes the Pareto frontier of reward and distribution preservation. As demonstrated in Table 3, GILC ranks first in the overall success rate, which explicitly balances these competing objectives. Furthermore, because GILC does not modify the underlying diffusion model's weights, the risk of reward hacking is substantially lower than fine-tuning approaches.
>
> ## Q4. Theoretical Intuition for this simplification.
>
> This simplification is inspired by recent advances in score distillation [5,6], where omitting the model’s Jacobian in image diffusion improves optimization stability.
>
> In addition, we provide an intuitive explanation here to help reviewers understand why GILC can still work even when it ignores the model Jacobian. Let $p(z_s|z_t) = q(z_s|z_t, x_\theta)$ denote the transition kernel, where $x_\theta = \text{softmax}(\eta) = \mathbb{E}[x|z_t]$. Introducing a reward $r(x)$ yields a reward-tilted target $\mathbb{E}[x|z_t, r]$.  The GILC update approximates this quantity by:
> $$
> \mathbb{E}\left[x \mid z_t, r\right] \approx \operatorname{softmax}\left(\eta + \frac{1}{\beta} \nabla_\eta \mathbb{E}[r(x)]\right)
> $$
> Thus, the simplified guidance term in Eq. (14) provides a first-order approximation of the reward-conditioned posterior expectation, shifting the model prediction from the unconditional estimator $\mathbb{E}[x|z_t]$ toward $\mathbb{E}[x|z_t, r]$, which effectively guides the transition kernel.
>
>
> ## Q5. Gradient comparation  of GILC-DB and GILC-PG.
>
> Qualitatively, both methods estimate the gradient but differ in trade-offs: GILC-DB uses the Straight-Through estimator. As shown in [4], it is biased but has low variance, leading to good performance with fewer Monte Carlo samples. GILC-PG uses policy gradients treating the reward as a black box. It is unbiased but typically exhibits high variance. Therefore, when the reward is differentiable, we recommend using GILC-DB for efficiency and stability.
>
> ## Q6. Ablation study of variance reduction techniques.
>
> We conducted an ablation study by disabling group normalization in DNA sequence generation and incorporating other state-of-the-art variance reduction techniques, including RLOO [7] and ReMax [8], using leave-one-out and greedy decoding as baselines.
> Our results indicate that group normalization plays a critical role in GILC-PG performance, and other variance reduction techniques also provide moderate improvements.
>
> |    | Pred-Activity | ATAC-Acc | 3-mer Corr | JASPAR Corr|
> |-----|-----|-----|-----|-----|
> | w/o group-normalisation | 3.77 | 34.8 | 0.304 | 0.596|
> | GRPO |	5.21 | 84.0 | **0.910** | **0.937**|
> | RLOO|	5.18 | 84.2 | 0.901 | 0.928|
> | ReMax|	**5.24** | **85.7** | 0.907 | 0.933|
> ***
> [1] *A general framework for inference-time scaling and steering of diffusion models. arXiv 2025.*
>
> [2] *Inference-time scaling of diffusion models through classical search. arXiv 2025.*
>
> [3] *Derivative-free guidance in continuous and discrete diffusion models. arXiv 2024.*
>
> [4] *Estimating or propagating gradients through stochastic neurons for conditional computation. arXiv 2013*
>
> [5] *Dreamfusion: Text-to-3d using 2d diffusion. ICLR 2023.*
>
> [6] *Delta denoising score. CVPR 2023.*
>
> [7] *Back to Basics: Revisiting REINFORCE-Style Optimization for Learning from Human Feedback in LLMs. Axiv 2024*
>
> [8] *ReMax: A Simple, Effective, and Efficient Reinforcement Learning Method for Aligning Large Language Models. ICML 2024.*

---

> > ### Author Rebuttal · Reviewer_LdoM · 2026-04-01
> >
> > I appreciate the authors’ clear and thorough responses. My main concerns have been adequately addressed. I will keep my score.

---

### Official Review · Reviewer_KoyG · 2026-03-03

**Soundness:** 2
**Presentation:** 3
**Significance:** 2
**Originality:** 2
**Overall Recommendation:** 4
**Confidence:** 5

**Summary:**

This paper introduces Gradient-Informed Logit Correction (GILC), a plug-and-play framework designed to guide discrete diffusion models. The main idea is to sample from the reward-tilted target distribution at the diffusion step. The framework accommodates both differentiable rewards (using Gumbel-Softmax trick) and non-differentiable rewards (via REINFORCE gradient estimator). Empirically, it demonstrates that GILC achieves promising performance in DNA sequence design and protein generation.

**Compliance With Llm Reviewing Policy:**

Affirmed.

**Final Justification:**

I increased my score from weak reject to weak accept, given the proposed method is simple and effective. However, some of my concerns still remain:

- The evaluation is limited. It would be better to show the result in text-to-image scenarios.

- The paper does not give correct credit to [1]. Specifically, while [1] does not conduct an experiment on non-differentiable rewards, it also discusses the REINFORCE trick in the appendix. However, the paper does not cite [1].

**Key Questions For Authors:**

- In Table 2, the `Num-of-Diff` for SVDD is listed as 20. Could you clarify the reasoning behind this? To my understanding, SVDD should only require 1 call to the denoising model per step, similar to the proposed  GILC method.
- Could you provide an ablation study detailing the impact of omitting the group-normalisation within the GILC-PG formulation?
- The logical flow in Section 3.2  could be improved. Currently, the conceptual link between the variational value estimation and the optimal reverse process is a bit difficult to follow. To clarify this, I recommend explicitly framing the section by stating that a Taylor approximation is utilised to approximate the reward-tilted target defined in Equation 6
- There is a missing reference citation at line 885

**Limitations:**

The paper's primary limitations are methodological and empirical. Methodologically, the approach shares significant overlap with previous work using Gumbel-Softmax to approximate the reward-tilted target, a connection that is currently unaddressed. Empirically, the evaluation is limited in scope and needs validation across more diverse modalities. Additionally, the paper is missing a rigorous theoretical analysis regarding the convergence guarantees of the proposed method.

**Strengths And Weaknesses:**

**Strongths**:

- The paper introduces a highly practical, training-free, plug-and-play framework (GILC) for guiding discrete diffusion models. The overall idea is straightforward and easy to apply.
- GILC demonstrates promising performance in scientific domains and outperforms competing training-free methods such as SVDD and SMC.

**Weakness**:

- Lack of Theoretical Convergence Guarantees: The paper does not provide a formal proof of convergence for the proposed method. Unlike baseline approaches such as SMC and SVDD, which offer asymptotic convergence guarantees, GILC appears to be a biased estimator. The theoretical limits and bounds of this bias are not rigorously established.
- Missing Discussion on Closely Related Work: There is a key missing reference [1] that must be addressed. The technique of utilizing Gumbel-Softmax for gradient estimation to approximate optimal proposals for SMC has been previously proposed in [1]. The core mechanism of GILC-DB shares significant conceptual similarities with this prior work, with the primary technical distinction being GILC-DB's deliberate omission of the Jacobian term.
- Limited Scale and Modality of Evaluation: The empirical evaluation is currently restricted to relatively small-scale scientific generation tasks (e.g., DNA sequences with a vocabulary size of 4 and a sequence length of 200). It would be highly beneficial to evaluate it on larger-scale discrete spaces, such as discrete image generation.

[1] Ou, Zijing, Chinmay Pani, and Yingzhen Li. "Inference-Time Scaling of Discrete Diffusion Models via Importance Weighting and Optimal Proposal Design." *arXiv preprint arXiv:2505.22524* (2025).

---

> ### Author Rebuttal · Authors · 2026-03-30
>
> Thank you for your constructive comments. Below is our response to your main concerns.
>
> ## Q1.Convergence Guarantees
>
> We agree that formal theoretical bounds are a valuable direction for future work, but we respectfully highlight that the asymptotic convergence guarantees of SVDD and SMC rely on assumptions, such as an **infinite number of particles** and **perfectly accurate value estimation**, that are rarely satisfied. Under realistic, finite-particle regimes, these baseline approaches also inevitably introduce estimation bias.﻿
>
>  Our primary objective with GILC is to provide a practical, computationally efficient implementation for the reward-tilted distribution defined in Eq. 6 of our manuscript. As demonstrated in Fig. 2(a), GILC exhibits substantially lower value estimation bias than SVDD and SMC in practice. This translates directly to the empirical results (Tables 1, 2, and 5), where GILC consistently navigates to higher-reward regions than competing training-free methods under the same inference budgets.
>
> ## Q2. Discussion on related work [1].
>
>  We were initially unaware of [1]. After reviewing it, we found that our work differs significantly from [1].
> While both works cleverly utilize the Gumbel-Softmax trick for gradient estimation to improve discrete diffusion, GILC diverges from [1] in three fundamental ways:
>
> 1. We derive our value function estimation directly from the variational objective, establishing a distinct theoretical insight for our approach.
>
>
> 2. A core mechanism of GILC is applying guidance directly to the logits, which we found to be critical for discrete diffusion. In contrast, [1] applies guidance to the noisy state $z_t$. As shown in Fig. 2(b), this logit-space guidance is the primary reason GILC outperforms fine-tuning baselines in guidance performance.
>
> 3. GILC-PG introduces an alternative formulation using group relative advantages, successfully extending our framework to non-differentiable rewards, a capability not explored in [1].
>
> Thank you for pointing this out. We will cite [1] and include this comparative discussion in the related work section.
>
> ## Q3. Evaluation of discrete image generation.
>
> To demonstrate the scalability of our framework, we have conducted new experiments on class-conditional generation using the CIFAR-10 discrete diffusion model provided in [2]. Both GILC-DB and GILC-PG successfully scale to this domain. Visualizations of the generated images, suggesting that GILC produces class-conditional images with good sample quality and confirming the efficacy of our method on larger discrete spaces.  The generated images are provided here: [https://anonymous.4open.science/r/GILC_cifar-4A76/](https://anonymous.4open.science/r/GILC_cifar-4A76/).
>
> ## Q4. Num-of-Diff for SVDD
>
> To select the optimal particle, SVDD must evaluate the value of each candidate particle sampled from the transition distribution. This value assessment requires computing the reward $r(x_\theta(z_s, s))$, which necessitates a separate forward pass through the denoising network for every candidate. Therefore, the Num-of-Diff for SVDD scales linearly with the number of particles (20 in this experimental setup), whereas GILC only requires 1 call to the denoising model per step.
>
>
> ## Q5. Ablation study of group-normalisation.
>
> We evaluated the impact of group-normalisation on the DNA sequence design task. As shown below, group-normalisation is a critical component; it leverages the relative quality among different samples to establish a stable guidance direction.
>
> |    | Pred-Activity| ATAC-Acc| 3-mer Corr |JASPAR Corr |
> |--------|--------|--------|--------|--------|
> | w/ group-normalisation | **5.21** | **84.0** | **0.910** | **0.937** |
> | w/o group-normalisation | 3.77 | 34.8 | 0.304 | 0.596 |
>
> ##  Q6. Logical flow and missing reference.
> We appreciate the helpful suggestions for improving Section 3.2. In the revision, we will explicitly frame this section by stating that a Taylor approximation is utilized to approximate the reward-tilted target defined in Eq. 6.  We will also add the missing citation at line 885.
>
>
>
> ***
> [1] *Inference-Time Scaling of Discrete Diffusion Models via Importance Weighting and Optimal Proposal Design. arXiv 2025.*
>
> [2] *A Continuous Time Framework for Discrete Denoising Models. NeurIPS 2022.*

---

> > ### Author Rebuttal · Reviewer_KoyG · 2026-04-04
> >
> > Thanks for the reply. Some of my concerns still remain.
> >
> > - It is very nice to see the image results in CIFAR10. However, it would be better to show the result on text-to-image following the setting in [1].
> > - While [1] does not conduct an experiment on non-differentiable rewards. [1] also discusses the REINFORCE trick in the appendix. It would be better to give credit to [1] in the main paper.
> >
> > Nevertheless, I think the proposed method is simple and effective. I will increase my score to weak accept.
> >
> > [1] Inference-Time Scaling of Discrete Diffusion Models via Importance Weighting and Optimal Proposal Design. arXiv 2025.

---

> > > ### Author Response · Authors · 2026-04-06
> > >
> > > Thank you very much for your thoughtful follow-up and for increasing your score!
> > >
> > > 1. Following your recommendation, we evaluated our method on a text-to-image setting using the Meissonic [2] model with Aesthetic Score as the reward. The results show that GILC remains effective even for high-resolution text-to-image generation. We have added updated visualization results at the link: [https://anonymous.4open.science/r/GILC_cifar-4A76/](https://anonymous.4open.science/r/GILC_cifar-4A76/).
> > >
> > > 2. Thank you also for pointing out the discussion of [1].  We will ensure that we credit [1] in the main paper and clarify both the relationship and the differences between their method and ours.
> > >
> > > Best regards.
> > > ***
> > >
> > > [1] *Inference-Time Scaling of Discrete Diffusion Models via Importance Weighting and Optimal Proposal Design. arXiv 2025.*
> > >
> > > [2] *Meissonic: Revitalizing masked generative transformers for efficient high-resolution text-to-image synthesis. ICLR 2025.*

---

### Official Review · Reviewer_RwCp · 2026-03-14

**Soundness:** 3
**Presentation:** 4
**Significance:** 3
**Originality:** 3
**Overall Recommendation:** 5
**Confidence:** 4

**Summary:**

This paper presents GILC, a training-free plug-and-play framework for guiding discrete diffusion models. The key insight is that the pre-trained denoising network serves as a variational proxy for value function estimation (Eq. 9-11), avoiding auxiliary training. The paper identifies Jacobian instability in discrete diffusion, i.e., with a large condition number, and proposes Jacobian-free logit correction. Based on the observations, the authors propose two variants: GILC-DB with direct backprop for differentiable rewards, and GILC-PG with policy gradient for non-differentiable rewards. The proposed methods are evaluated on DNA enhancer design, protein inverse folding, and molecular generation, outperforming baseline methods.

**Compliance With Llm Reviewing Policy:**

Affirmed.

**Key Questions For Authors:**

- The insight of model Jacobian being numerically unstable is intriguing. Have the authors conducted an ablation study comparing logit correction to the full gradient of the value function?
- In the Gumbel-softmax reparameterization, how does the choice of the softmax temperature affect the final performance?

**Limitations:**

Yes.

**Strengths And Weaknesses:**

### Strengths
- The idea of using MC rollouts as a variational proxy for value function estimation is well formulated.
- The Jacobian instability analysis (Figure 3) is thorough and convincing. The proposed logit correction bypass is intuitive and effective. The proof of equivalence in Appendix B.2 is also satisfactory.
- Strong empirical results: GILC has a strong performance on DNA design, inverse protein folding, and multimodal molecule generation tasks, outperforming existing guidance methods such as SMC, SVDD, and TFG-Flow.
- This paper is well written. The logical progression of this paper is coherent and easy to follow.

### Weaknesses

- Missing comparison or conceptual discussion with more recent guidance method [1] and finetuning method [2] for discrete diffusion models. Both of them report results on the same DNA design benchmark.
- The authors claim GILC-PG as a solution to non-differentiable reward functions, but it seems all tasks considered are differentiable. The results could potentially be improved by considering a non-differentiable reward function.

[1] Chu et al. "Split Gibbs Discrete Diffusion Posterior Sampling." NeurIPS 2025.

[2] Zekri et al. "Fine-Tuning Discrete Diffusion Models with Policy Gradient Methods." NeurIPS 2025.

---

> ### Author Rebuttal · Authors · 2026-03-29
>
> Thanks for your constructive feedback and valuable suggestions. Below, we address each comment in detail.
>
> ## Q1. Comparison and discussion with [1] and [2].
>
> Thank you for pointing that out. In fact, we have already cited and discussed [1] and [2] in Appendix A.1 of our manuscript.
>
> [1] uses variable splitting to implement discrete diffusion condition sampling. However, as noted in its limitations section, this method is only applicable to diffusion with a uniform transition kernel. It cannot be applied to diffusion with a masked transition kernel, which is more common. Our method does not have this limitation. Given the differences in the base models used, a quantitative comparison may not be fair.
>
> [2] is a method that uses policy gradient to fine-tune discrete diffusion, and we have conducted a comparison. As shown below, we find that the proposed GILC can produce comparable results without the need for training.
>
> | Method  | Pred-Activity | ATAC-Acc | 3-mer Corr | JASPAR Corr |
> |---------|--------------|----------|------------|-------------|
> | SEPO    | **7.55**         | **99.5**     | 0.500      | 0.649       |
> | GILC-DB | 7.04         | 95.2     | **0.900**      | **0.935**       |
>
>
> ## Q2. Non-differentiable reward functions.
>
> Thank you for your suggestion. Here, we incorporate a structure-guided molecular generation task on the QM9 dataset, in which structures are characterized using molecular fingerprints [3]. A molecular fingerprint consists of a series of bits that indicate the presence or absence of specific substructures within a molecule. This process is non-differentiable. We use the Tanimoto coefficient [4] to measure the structural similarity. The results are based on an average of three random seeds, and we found that GILC-PG outperforms the baselines.
>
> | Method | Similarity |
> |--------|------------|
> | Best-of -n | 0.182 |
> | SMC | 0.178 |
> | SVDD | 0.234 |
> |TFG-Flow|0.271|
> |GILC-PG|**0.308**|
>
> ## Q3. Ablation study comparing logit correction to the full gradient of the value function.
>
> Due to space constraints in the main text, the ablation results for the logit-space correction can be found in Tables 6 and 7 in the Appendix; the results indicate that the logit-space correction significantly improves guidance performance.
>
> ## Q4. Effect of softmax temperature.
>
> The experiment in our manuscript used the default temperature of $\tau=$1. We conducted experiments on 0.5, 1.0, and 1.5 and found that these temperature ranges worked well. Generally speaking, overly small temperatures may increase gradient variance and amplify estimation errors, while excessively large temperatures may oversmooth the categorical distribution and weaken guidance strength.
>
> | |$\tau = 0.5$| $\tau = 1.0$ |$\tau = 1.5$ |
> |--------|--------|------------|------------|
> |Pred-ddG|1.247|1.430|1.510|
> |scRMSD| 0.973| 0.968| 0.940|
> |Success Rate (%)| 81.7| 82.4| 82.2|
>
>
>
>
>
>
>
>
>
> ***
>
> [1]  *Split Gibbs Discrete Diffusion Posterior Sampling. NeurIPS 2025.*
>
> [2] *Fine-Tuning Discrete Diffusion Models with Policy Gradient Methods. NeurIPS 2025.*
>
> [3] *Inverse design of 3d molecular structures with conditional generative neural networks. Nature Communications.*
>
> [4] *Why is tanimoto index an appropriate choice for fingerprint-based similarity calculations? Journal of cheminformatics.*

---

> > ### Author Rebuttal · Reviewer_RwCp · 2026-04-06
> >
> > I would like to thank the authors for their detailed response. My concerns have been fully addressed, and my assessment of this paper has not changed.

---

### Official Review · Reviewer_d74n · 2026-03-16

**Soundness:** 1
**Presentation:** 2
**Significance:** 4
**Originality:** 3
**Overall Recommendation:** 4
**Confidence:** 3

**Summary:**

This paper introduces a training-free guidance method for discrete diffusion models. The approach uses the pretrained denoiser as a proxy distribution for reward estimation and then steers sampling by applying a correction to the model’s clean-prediction logits, with variants for both differentiable and non-differentiable rewards. The method is evaluated on DNA, protein, and molecular generation tasks, where it is reported to outperform prior training-free baselines while remaining computationally efficient.

**Compliance With Llm Reviewing Policy:**

Affirmed.

**Final Justification:**

The final rebuttal improves my assessment, and I am raising my score accordingly. In particular, the authors now clarify that the motivating gradient should be understood through a differentiable surrogate $v_\phi$, which addresses part of my original concern. That said, I still remain somewhat concerned that the paper presents parts of the method more formally than is warranted, especially since the manuscript’s original derivation is stated in terms of $v$ from Eq. 7 and the implemented method ultimately relies on additional heuristic approximations. The method itself is very interesting, and I think trying to present it as fully principled somewhat detracts from what is otherwise an original and compelling contribution. Overall, I now view the paper as a promising and technically interesting contribution, but I still encourage the authors to more clearly distinguish the principled components from the heuristic ones in the final version.

**Key Questions For Authors:**

1) What is the precise mathematical object denoted by $\nabla_{z_t} v(z_t)$ in Eq. 8, given that $z_t$ is discrete?

2) What is the justification for approximating $\nabla_{z_t} v(z_t)$ if the Taylor-expansion argument in Sec. 3.1 does not hold in the discrete setting?

3) What exactly justifies the transition from the chain-rule formulation in Sec. 3.3 / Eq. 13 to the implemented logit-space update in Eq. 14?

**Limitations:**

Yes

**Strengths And Weaknesses:**

Significance. The paper addresses an important problem in training-free guidance for discrete diffusion, and the empirical results are strong across several tasks.

Originality. The core idea is interesting. The method uses the pretrained denoiser as a variational proxy and then guides sampling through a correction in logit space rather than state space. I am not deeply familiar with the full literature, but as far as I can tell, there are novel ideas here.

Soundness. My main concern is with the paper’s core motivation in Sec. 3.1. The derivation is built around the gradient of the value function with respect to the discrete noisy state in Eq. 8, but this gradient is not mathematically well defined in a discrete state space. As written, there is no literal underlying object here for the method to estimate, so the Taylor expansion argument does not go through in the stated form. This is not just an approximation issue. It undermines the claimed motivation for the method. Even setting aside the issue that the target gradient is not well defined, the variational proxy itself is yet another approximation to the original objective rather than a direct derivation from it. Beyond this, the method later drops the model Jacobian and instead applies the logit space update in Eq. 14, again without a rigorous justification. Thus, the implemented algorithm is better understood as a practical heuristic than as a rigorous discrete state gradient method. A similar issue appears in Sec. 3.4.

Presentation. I found the presentation confusing because the paper moves between rigorous and heuristic language without clearly distinguishing the two. The method is sometimes presented as a principled gradient derivation, but key steps are later replaced by surrogate updates without being clearly flagged as such. This is especially problematic because the motivating value gradient is not well defined in the discrete setting to begin with. For instance, the statement below Eq. 13 that “this formulation is mathematically rigorous” is confusing and, as written, not correct. Similarly, the claim in Eq. 8 that the quantity can be computed with a single forward and backward pass seems to conflate a mathematical gradient with a neural network surrogate computation. The paper would be clearer and more honest if it presented the method consistently as a heuristic rather than a rigorous gradient-based derivation.

---

> ### Author Rebuttal · Authors · 2026-03-27
>
> Thank you for recognizing the originality of our work and for your feedback. Below is our response to your concerns.
>
> ## Q1. Justification of $\nabla_{z_t}v(z_t)$ and Taylor expansion.
>
> **Precise mathematical object of $\nabla_{z_t}v(z_t)$.** While gradients are undefined for discrete variables in the strict sense, we can treat $z_t$ as a continuous one-hot tensor, allowing a surrogate gradient  $\nabla_{z_t}v(z_t)$ to be computed. This is common practice: prior work (e.g., [1], Fig. 1) shows that gradients over discrete representations can effectively guide categorical sampling, and we adopt the same convention.
>
> **Taylor expansion argument.** The basis for this Taylor expansion also draws on recent work on discrete diffusion (e.g., Eq. 12 in [2], Sec. 5 in [3]). This Taylor approximation has demonstrated strong empirical performance in previous works. Our method follows the same principles and inherits these advantages.
>
> ## Q2. Justification of transition from Eq. 13 to  Eq. 14.
>
>
> **Design motivation.**  As indicated by the title of Section 3.3, it is a practical design choice to improve numerical stability. As shown in Fig. 3, the discrete diffusion Jacobian exhibits a poor condition number, which may hinder effective value guidance. Our transformation is inspired by analogous techniques in continuous diffusion (e.g., [4, 5,6]), where omitting the model Jacobian term has been shown to improve gradient optimization. We adopt a similar spirit for discrete diffusion.
> ﻿
>
> Beyond numerical motivation, Eq. (14) admits a useful probabilistic interpretation. Let $p(z_s|z_t) = q(z_s|z_t, x_\theta)$ denote the diffusion transition kernel, depends on the expectation $x_\theta = \text{softmax}(\eta) = \mathbb{E}[x|z_t]$.After introducing a reward $r(x)$, the ideal guidance target becomes the reward-tilted expectation $\mathbb{E}[x\mid z_t,r]$. Our update approximates this quantity via:
> $$
> \mathbb{E}\left[x \mid z_t, r\right] \approx \operatorname{softmax}\left(\eta + \frac{1}{\beta} \nabla_\eta \mathbb{E}[r(x)]\right)
> $$
> Thus, the simplified guidance term in Eq. (14) provides a first-order approximation of the reward-conditioned posterior expectation $\mathbb{E}[x|z_t, r]$, which can guide the transition kernel $q(z_s|z_t, x_\theta)$ toward the high reward distribution.
> ﻿
> ## Q3.  On the variational proxy.
>
> We agree that the variational surrogate introduces an additional level of approximation. However, computing $v(z_t)$ exactly requires evaluating $p(x|z_t)$, which is intractable in general. **Variational inference is a standard and well-accepted technique for handling such intractable distributions.**
> ﻿
> Crucially, as shown in Fig. 2(a) of our manuscript, our method substantially outperforms the current best value estimation approach. We think that the approximation is well-justified within the plug-and-play framework, where some degree of tractability trade-off is inherent and widely accepted in the literature (similar to Eq.4 in [7]).
> ﻿
> ## Q4. Regarding the presentation.
>
>
> **Presentation of Eq. 13 and Eq. 8.** As clarified in our response to Q1, gradients in Eq. 13 are defined under the continuous one-hot variables, as in [1]. Similarly, the claim of Eq. 8  using a single forward–backward pass follows the claim of Eq. 12 in [2].
> ﻿
> More broadly, we would like to clarify that our method follows the common structure shared by many diffusion guidance approaches (e.g., [7,8,9]), rather than being completely heuristic:
> ﻿
> 1. Start from a principled gradient-based objective;
> 2. introduce tractable approximations where exact computation is infeasible;
> 3. retain compatibility with pretrained diffusion models in a plug-and-play manner.
> ﻿
> We appreciate the reviewer‘s feedback. In the revised version, we will provide additional explanations to further enhance the design insights of our framework.
>
> ***
> [1] *Oops i took a gradient: Scalable sampling for discrete distributions. ICML 2021.*
>
> [2] *Simple guidance mechanisms for discrete diffusion models. ICLR 2025.*
>
> [3] *Digress: Discrete denoising diffusion for graph generation.ICLR 2023.*
>
> [4] *Dreamfusion: Text-to-3d using 2d diffusion. ICLR 2023.*
>
> [5] *Delta denoising score. CVPR 2023.*
>
> [6] *Direct Diffusion Bridge using Data Consistency for Inverse Problems. NeurIPS 2023*
>
> [7] *Pseudoinverse-guided diffusion models for inverse problems. ICLR 2023*
>
> [8] *Diffusion posterior sampling for general noisy inverse problems. ICLR 2023*
>
> [9] *FreeDoM: Training-Free Energy-Guided Conditional Diffusion Model. ICCV 2023*

---

> > ### Author Rebuttal · Reviewer_d74n · 2026-04-01
> >
> > Thank you for the response. Unfortunately, I remain unconvinced by the justification for the gradient in Eq. 8 and the associated Taylor expansion.
> >
> > My concern is not with the general use of continuous relaxations in discrete problems. That can be valid when one begins with a genuinely differentiable function $v : \mathbb{R}^D \to \mathbb{R}$, and then restricts the domain of that function to a discrete subset of $\mathbb{R}^D$. In that setting, the gradient $\nabla v(x)$ is a well-defined mathematical object, and a first-order Taylor expansion can legitimately motivate an approximation to differences $v(x') - v(x)$ between nearby discrete points.
> >
> > This is why, as I understand them, the arguments in [1] and [2] are different from what is done here. In [1], the relevant object is a differentiable function $f$ on a continuous space, and the method applies the approximation after restricting the domain to discrete configurations. In [2], the relevant object is likewise a differentiable function, namely a neural-network quantity such as $\log p_\phi$, so again the Taylor argument is applied to a function that is genuinely differentiable before its domain is restricted to discrete points. In both cases, there is a true underlying differentiable function first, and only then a restriction of its domain to a discrete set. That is what makes the gradient and Taylor approximation mathematically meaningful there.
> >
> > In your paper, however, $v$ is introduced as a value function on discrete states $z_t$. As written, I still do not see what the underlying differentiable function is supposed to be. If $z$ is not literally a token, sequence, or other discrete diffusion state, then what exactly is $v(z)$? What is its domain? What is the function being differentiated? If the authors cannot answer these questions precisely, then I do not see how $\nabla_{z_t} v(z_t)$ can be regarded as a defined mathematical object.
> >
> > Put differently, representing $z_t$ as a continuous one-hot vector does not by itself solve the problem. That only changes the representation of the input. It does not define a differentiable extension of the value function $v$ off the discrete state space. Without such an extension, I do not think the Taylor expansion in Sec. 3.1 is justified as written.
> >
> > To be clear, I am not objecting to every use of this type of continuous relaxation in the paper. For example, when you differentiate the reward $r(x)$, that part is valid, because there the underlying object really is a differentiable function of its input. In that case, applying the same kind of approximation used in [1] and [2] makes sense. My concern is specifically with the separate claim that $\nabla_{z_t} v(z_t)$ is likewise available in Eq. 8. Those are not the same thing.
> >
> > This is also why I remain unconvinced by the response that appeals to prior work and empirical effectiveness. Even if similar approximations have worked well elsewhere, that does not answer the mathematical question here unless one can identify the underlying differentiable function whose domain has been restricted to the discrete states. As written, I do not see that for $v$.
> >
> > The comparison to [7], [8], and [9] also does not resolve this issue, because those works concern continuous diffusion models, where the relevant underlying objects are defined on continuous spaces and are differentiable, so their use does not by itself justify the same step for a value function defined only on discrete states.
> >
> > Because the entire motivation in Sec. 3.1 hinges on approximating $\nabla v$, I do not view this as a small technical issue or a minor presentation issue. It affects the central derivation used to motivate the method.
> >
> > Because this issue sits at the core of the paper’s motivation, I do not feel able to raise my score at this stage. If the authors can clearly define the underlying differentiable function whose restriction yields the discrete value function they use, and rewrite the presentation accordingly, I would be happy to reconsider.

---

> > > ### Author Response · Authors · 2026-04-02
> > >
> > > Thank you for your additional feedback. We completely agree with your assessment: a Taylor expansion is only mathematically valid if applied to a continuous, differentiable function whose domain is subsequently restricted to a discrete subset, exactly as in [1] and [2]. To address this issue precisely, we clarify below the intended construction of the value function used in Eq. 8 and revise the presentation of Sec. 3.1 accordingly.
> > >
> > > The underlying differentiable function we refer to is, in fact, a parameterized continuous neural network $v_\phi(\cdot) : \mathbb{R}^{K \times L} \to \mathbb{R}$. In standard discrete guidance approaches [2], this network can be explicitly trained via regression to fit the target reward:
> > > $$
> > > \min_\phi  \mathbb{E}\_{p(z_t|x)}\|v_\phi(z_t)-r(x)\|^2
> > > $$
> > > where the optimal solution is equal to the definition of value function in our manuscript, i.e., $v_\phi(z_t) \approx \mathbb{E}\_{p(x|z_t)}[r(x)]$. Because $v_\phi$ operates on continuous representations , its gradient $\nabla_{z_t} v_\phi(z_t)$ is a well-defined mathematical object. By treating the states $z_s$ as points restricted to the discrete domain of this function, the first-order Taylor expansion becomes rigorous and justified.
> > >
> > > Our work builds precisely on this premise. However, instead of training a separate value network $v_\phi$ from scratch for each new target (which is computationally expensive), our core contribution is proposing a plug-and-play method that estimates this continuous value gradient using an off-the-shelf differentiable diffusion network.
> > >
> > > Here, we propose the following improvement to the presentation in Eq. 8 of the manuscript (**lines 200–218, left**).
> > >
> > > > "For a sequence of length $L$, computing Eq. 6 directly involves a computationally intractable term over a state space of size $K^L$. Similar to existing discrete guidance methods [2], one can address this by learning a differentiable network $v_\phi$ via regression: $$\min_\phi  \mathbb{E}\_{p(z_t|x)}\|v_\phi(z_t)-r(x)\|^2$$
> > >
> > > > Crucially, $v_\phi(\cdot): \mathbb{R}^{K \times L} \rightarrow \mathbb{R}$ is a genuinely differentiable continuous function. By treating the discrete states $z_s$ as points restricted to the continuous domain of this function (e.g., via continuous one-hot representations), we can legitimately apply a first-order Taylor expansion around $z_t$:
> > > > $$v_\phi(z_s) \approx v_\phi(z_t) + \langle \nabla_{z_t} v_\phi(z_t), z_s - z_t \rangle \quad (8)$$
> > > > Under this formulation, estimating the optimal policy hinges entirely on calculating the value function gradient, $\nabla_{z_t} v_\phi(z_t)$, which can be calculated efficiently with
> > > a single forward and backward pass of $v_\phi(\cdot)$. While sound, this standard approach requires retraining the value network $v_\phi$ for every new reward target. In the following section, we propose a plug-and-play alternative that bypasses this retraining. Specifically, we demonstrate how to estimate this value gradient directly using an off-the-shelf differentiable diffusion network. In our subsequent derivations, we abbreviate the differentiable value function $v_\phi$ as $v$ and omit the subscript for brevity."
> > >
> > > We hope this clarification will resolve any ambiguity. We will revise the manuscript accordingly and sincerely hope that the reviewer can reconsider the evaluation of our manuscript.

---

### Decision · Program_Chairs · 2026-04-30

**Decision:**

Accept (regular)

**Comment:**

This paper proposes Gradient-Informed Logit Correction (GILC), a training-free, plug-and-play guidance method for discrete diffusion models. The approach leverages a pretrained denoiser as a variational proxy and applies a Jacobian-free logit correction to guide sampling. It supports both differentiable and non-differentiable rewards and demonstrates strong empirical performance on multiple sequence generation tasks.

All reviewers agree that the paper addresses an important problem and that the proposed method is practical, simple, and empirically effective, outperforming prior training-free baselines. The work is viewed as a useful contribution with potential impact.

However, reviewers also raise concerns about the theoretical grounding and positioning. In particular, the derivation in Section 3.1 relies on gradients over discrete variables without a clearly defined underlying differentiable function, and this issue is not fully resolved in the rebuttal. Additional concerns include missing related work, limited evaluation scope, and some imprecision in presentation. Overall, the method appears better characterized as a principled but heuristic approach rather than a fully rigorous derivation.

Taking these points together, while the theoretical justification is not fully satisfactory, the empirical strength and practical relevance of the method justify acceptance. The paper would benefit from clarifying its assumptions, better distinguishing heuristic components, and improving positioning with respect to prior work.